# CARE: Adaptive Calibration for Reliable Recommendations

**Nitin Bisht** [1,2]   **Huan Huo** [1]   **Xiuwen Gong** [3]   **Guandong Xu** [2]

## Abstract

Modern recommender systems are typically trained offline and deployed with parameters held fixed between periodic refreshes, yet user behavior can evolve substantially during deployment. This can cause ranking utility to degrade over time and makes it difficult to provide formal guarantees about recommendation quality. We propose **CARE**, an adaptive calibration framework that wraps an arbitrary backbone recommender and outputs variable-size recommendation sets with finite-sample performance guarantees over interaction streams. CARE combines (i) a loss-based monitoring module that localizes behavioral changes and triggers threshold recalibration, and (ii) an online aggregation rule that promotes compact recommendation sets by dynamically reweighting candidate set predictors. We provide theoretical results establishing finite-sample guarantees for utility-based risk control and bounds on the expected set size relative to the best constituent predictor. Experiments across multiple datasets and backbone models demonstrate that CARE improves robustness and maintains compact recommendation sets while preserving the desired statistical guarantees. The code and implementation are available in `https://github.com/kalpiree/CARE`.

## 1. Introduction

Recommender systems are central to platforms such as e-commerce, streaming, and location-based services (Hussien et al., 2021; Chang et al., 2017; Rohilla et al., 2021). While much effort has gone into improving pointwise accuracy via different backbone architectures (e.g., SASRec (Kang & McAuley, 2018), Caser (Tang & Wang, 2018), and FMLP-

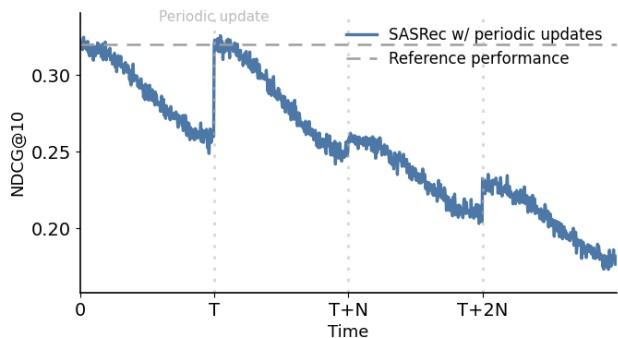

*Figure 1.* NDCG@10 over time on Gowalla Dataset using SASRec with periodic refreshes. Each block aggregates 25 interactions/user. Vertical dashed lines indicate refreshes every N blocks; the horizontal dashed line is the $t = 0$ reference performance.

Rec (Zhou et al., 2022)), where they are trained on historical interaction and deployed in canonical offline pipelines with parameters fixed between refreshes (Eskandanian & Mobasher, 2018; Chen et al., 2023), comparatively less attention has been paid to providing statistical guarantees on recommendation quality as user behavior can change between updates. Periodic retraining (Shen & Kurshan, 2023; Zhang et al., 2020) can partially restore performance, but, as illustrated in Figure 1, the utility (e.g., NDCG) of a periodically updated recommender on the Gowalla Dataset can still drop within each interval. We therefore lack a formal mechanism to provide statistical guarantees that the recommendation utility metric (for e.g. reference NDCG, Recall etc. performance) satisfies a user-specified utility-risk level under such updates.

As a result, we are motivated to develop a model agnostic, statistically principled recommendation calibration framework that provides finite-sample guarantees on recommendation quality under non-stationary user behavior in interaction streams. Specifically, we construct time-adaptive and compact recommendation sets that (i) adapt as the model's score distribution changes over time and (ii) control a performance-based risk criterion with high confidence (e.g., $1 - \delta = 0.95$) over time horizon.

Inspired by finite sample uncertainty quantification methods (Vovk et al., 2005; Angelopoulos & Bates, 2021; Bates et al., 2021), we propose **CARE** (Adaptive Calibration for

---

[1]University of Technology, Sydney [2]The Education University of Hong Kong [3]School of Computer Science, Wuhan University, Wuhan, China. Correspondence to: Xiuwen Gong <gongxiuwen@gmail.com>, Guandong Xu <gdxu@eduhk.hk>.

*Proceedings of the $43^{rd}$ International Conference on Machine Learning*, Seoul, South Korea. PMLR 306, 2026. Copyright 2026 by the author(s).

Reliable Recommendations), a plug-and-play framework for constructing recommendation sets with finite-sample, high-probability control of a utility-based risk over time. While uncertainty-based methods can be extended to temporal streams via adaptive updates (Xu & Xie, 2021; Xu et al., 2024), existing approaches typically rely on a fixed adaptation timescale (e.g., a rolling window or update rate), which can be brittle when the rate of preference change varies across users and time. Moreover, calibration signals in recommendation are temporally dependent and often noisy or heavy-tailed (Zaffran et al., 2022) (a challenge also reflected in general learning (Yu et al., 2023)), leading to unstable thresholds and resulting in overly conservative sets when calibration samples are small or unrepresentative.

As a result, existing formulations in their natural form do not directly address our key objectives. Specifically: (1) how to adapt calibration to user-specific, non-stationary preference changes without relying on a fixed rolling-window hyperparameter; (2) how to maintain compact and stable recommendation sets under noisy scores and heavy-tailed residuals; and (3) how to thereby provide finite-sample, high-probability guarantees over a finite deployment horizon on a utility-based risk criterion when calibration is temporally dependent.

To address these challenges, CARE, instead of relying on a fixed rolling window, firstly adapts calibration to user-specific, non-stationary preference changes by monitoring a loss-based change indicator, selecting an adaptive calibration segment, and updating the threshold accordingly. Next, to mitigate noisy scores and heavy-tailed residuals, it uses multiple base models trained offline to construct candidate prediction sets and aggregates them into a single recommendation set, stabilizing recommendations under score instability. Finally, to maintain compactness while preserving statistical control, it updates aggregation weights online via a size-regularized multiplicative-weights scheme that favors models producing smaller sets; together with the adaptive recalibration step, which provides finite-sample control of a utility-based risk criterion with at least $(1 - \delta)$ confidence over time. We also provide a theoretical analysis that quantifies both the utility risk-control guarantee and the induced set-size stability under the proposed aggregation and recalibration procedures. The framework is illustrated in Figure 2.

Our contributions are as follows:

- We formulate recommendation on interaction streams from the perspective of an uncertainty-aware set prediction task, and propose CARE, a reliable and adaptive framework that generates compact yet valid prediction sets with user-specified $\alpha$-risk under evolving (non-stationary) user behavior.
- We develop Dynamically Adaptive Uncertainty-aware

Optimization (DAUO), an efficient online calibration procedure that jointly updates aggregation weights and risk thresholds to balance set compactness and risk coverage, thereby achieving the objectives of CARE.
- Technically, we introduce a scalar loss-based non-stationarity metric that combines a relative loss-discrepancy term with a concept-sensitive divergence to quantify behavior change, enabling dynamic segmentation and localized threshold recalibration.
- Theoretically, we establish finite-sample guarantees for CARE. Specifically, we show that (1) expected size of aggregated prediction set is controlled relative to the best individual model at each timestamp (Theorem 1); and (2) expected utility-based risk at inference remains within a provable margin of $\alpha$ with probability at least $1 - \delta$ under evolving user behavior (Theorem 2).
- Empirically, we conduct extensive experiments across multiple backbone recommenders and benchmark datasets. We compare CARE to adaptation-aware recommendation baselines on standard ranking utility, and to conformal style calibration baselines in terms of risk/coverage and set-size efficiency. Results in Section 6 demonstrate that CARE improves robustness and set compactness while maintaining statistical control, consistent with our theory.

## 2. Related Work

### 2.1. Recommendation on Interaction Streams

Time-ordered recommendation has evolved from Markov-chain and factorization models (Rendle et al., 2010; 2009) to deep sequence architectures such as GRU4Rec (Hidasi et al., 2015) and transformer-based backbones (e.g., SAS-Rec (Kang & McAuley, 2018) and BERT4Rec (Sun et al., 2019)). These models improve pointwise accuracy by exploiting temporal structure, and a number of methods incorporate additional temporal or robustness mechanisms (e.g., disentanglement/self-supervision, time-aware attention, or causal components) (Ma et al., 2020; Li et al., 2020; Wang et al., 2023; Pan et al., 2024). However, they are trained offline and updated intermittently, without finite-sample guarantees that utility remains above a target *between updates* under deployment drift—motivating a separate reliability layer for recommendation decisions. More recently, variational and stochastic sequence models (Fan et al., 2021; Fang et al., 2020; Wang et al., 2022) have explored uncertainty-aware recommendation. While effective for robustness and calibration in practice, these approaches do not provide finite-sample guarantees on recommendation utility under evolving behavior in interaction streams.

## 2.2. Conformal Prediction

Conformal Prediction (CP) constructs prediction sets with finite-sample, distribution-free coverage under exchangeability by calibrating a nonconformity score on held-out data (Vovk et al., 2005; Shafer & Vovk, 2008; Romano et al., 2019; Angelopoulos & Bates, 2021). Split/inductive CP variants train a predictor on a training split and use a separate calibration split to select a threshold, yielding marginal coverage guarantees without requiring a probabilistic model. To remain valid under distribution shift in time-ordered data, online and adaptive CP methods update calibration statistics over time, e.g., using sliding windows or adaptive threshold updates (Gibbs & Candes, 2021; Zaffran et al., 2022; Xu et al., 2024; Liang et al., 2025; Angelopoulos et al., 2024; Wu et al., 2025). CP has also been explored for top-$N$ recommendation (Kagita et al., 2022; 2023; Bisht et al., 2025a;b). Recent work has further extended this direction to retrieval-augmented sequential recommendation (Zhang et al., 2025). However, existing formulations either focus on static calibration and do not directly target preference drift in interaction streams together with utility-based objectives and temporally dependent feedback signals.

## 3. Preliminaries

We first introduce the notations used in this paper. We consider $m$ users and n items represented by $\mathcal{U} = \{u_k\}_{k=1}^m$, and $\mathcal{I} = \{i_k\}_{k=1}^n$. For brevity, we use $u$ and $i$ to denote a user and an item in this paper. In a interaction streaming setting, every user $u$ has a chronological sequence of interacted items, denoted as $\mathcal{H}_u = [i^1, i^2, \ldots, i^{|T_u|}]$ where $i^t \in \mathcal{I}$ represents an item interacted with by user $u$ at time step $t$, and $|T_u|$ denotes the length of sequence for user $u$. The objective is, given a historical interaction sequence $\mathcal{H}_u$ for each user $u$, to predict the next item they are likely to interact with. Specifically:

$$i^{t+1} = \arg\max_{i \in \mathcal{I}} \mathcal{M}(i \mid \mathcal{H}_u), \qquad (1)$$

where, $\mathcal{M}(i \mid \mathcal{H}_u) : \mathcal{I} \times \mathcal{H}_u \rightarrow [0, 1]$, denotes the underlying recommender model. Given the nonstationarity of user preferences, however, there is no guarantee of the model's performance. This limitation motivates us to explore the creation of dynamic recommendation sets that adapt to changing user preferences, which we discuss next.

## 4. The Proposed Framework

In this section, we propose CARE, a novel framework that provides compact recommendations that adapt to evolving user behavior, with theoretical performance guarantees in an interaction-stream setting. We begin by defining the construction of the dynamic prediction set $\mathcal{C}^{t+1} \subseteq \mathcal{I}$ for a single underlying model, which is guided by a timestep-dependent threshold parameter $\lambda^t \in \Lambda \subset \mathbb{R}$. Specifically:

$$\mathcal{C}_{\lambda^t}^{t+1}(\mathcal{H}_u) = \big\{ i \in \mathcal{I} \mid \mathcal{M}(i \mid \mathcal{H}_u) \geq \lambda^t \big\}. \qquad (2)$$

For brevity we will refer to $\mathcal{C}_{\lambda^t} := \mathcal{C}_{\lambda^t}^{t+1}(\mathcal{H}_u)$. Our goal is, given the user-defined error rate $\alpha \in [0, 1]$, for every timestamp, the recommendations created ensure:

$$R(\mathcal{C}_{\lambda^t}) \leq \alpha. \qquad (3)$$

The risk $R(\cdot)$ in Equation (3) is defined as:

$$R(\mathcal{C}_{\lambda^t}) = \mathbb{E}_{u \sim \mathcal{U}}[\mathcal{L}_u(\mathcal{C}_{\lambda^t})], \qquad (4)$$

where $\mathcal{L}_u(\cdot)$ is the bounded user utility-based loss function defined as:

$$\mathcal{L}_u(\mathcal{C}_{\lambda^t}) = 1 - U_{metric}\big(i_{rel}^{t+1}, \mathcal{C}_{\lambda^t}\big). \qquad (5)$$

Here, $U_{metric}(\cdot)$ represents a generalized recommendation metric (such as Recall or NDCG) that measures performance of the recommendation set $\mathcal{C}_{\lambda^t}$ for any user $u$ given the relevant item $i_{rel}^{t+1}$.

## 4.1. Compact Set Construction

The threshold $\lambda^t$ in Equation (2) is learned from model scores, which are highly sensitive to the quality of the underlying recommender. When models are trained on sparse user histories, as is common in RS (Bertin-Mahieux et al., 2011; Cho et al., 2011), the resulting scores can become unstable, often leading to heavy-tailed residual distributions, which in turn destabilize threshold estimation and might result in overly conservative prediction sets.

To address this, we propose using a collection of $L$ base models:

$$\mathbb{M} = \Big\{ \mathcal{M}^1, \mathcal{M}^2, \ldots, \mathcal{M}^L \Big\}, \qquad (6)$$

where $L$ is the number of base models and each $\mathcal{M}^\ell$ is obtained, for example, by training on a bootstrap sample of the full user set $\mathcal{U}$, i.e., $\mathcal{U}^\ell \subseteq \mathcal{U}$.

Firstly, for each model $\mathcal{M}^\ell$, we generate a prediction set $\mathcal{C}_\ell^{t+1} := \mathcal{C}_{\lambda_\ell^t}^{t+1}(\mathcal{H}_u)$, guided by its own threshold $\lambda_\ell^t$ and targeting the per-model analogue of Equation (3). Next, we aggregate these sets, generated using $\boldsymbol{\lambda}^t = \{\lambda_\ell^t\}_{\ell=1}^L$, into an aggregated recommendation set. Specifically:

$$\mathcal{C}_{\boldsymbol{\lambda}^t}^{t+1,\text{agg}} = \mathcal{A}\Big(\{\mathcal{C}_\ell^{t+1}\}_{\ell \,:\, u \in \mathcal{U}^\ell}, \mathbf{w}^t\Big), \qquad (7)$$

where $\mathcal{A}(\cdot, \mathbf{w}^t)$ is an aggregation operator that merges the individual set predictors $\mathcal{C}_\ell^{t+1}$ using a weight distribution $\mathbf{w}^t \in \Delta^L$, the $(L-1)$-dimensional probability simplex (i.e., $\Delta^L = \{w \in \mathbb{R}^L : w_\ell^t \geq 0, \sum_{\ell=1}^L w_\ell^t = 1\}$), where each $w_\ell^t$ determines the contribution of model $\mathcal{M}^\ell$ at time $t$.

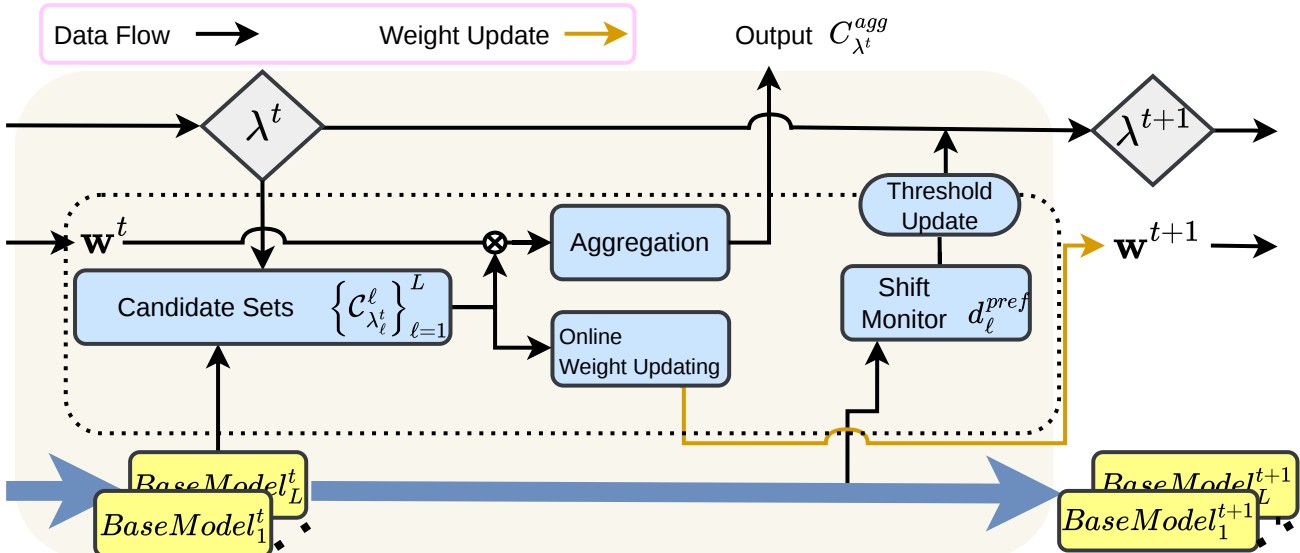

*Figure 2.* **Overview of the CARE Framework**. The candidate sets from multiple experts are aggregated into a recommendation set; a non-stationarity monitor triggers recalibration; online weighting promotes compact sets while preserving risk control.

The aggregation operator $\mathcal{A}(\cdot, \mathbf{w}^t)$, following (Gasparin & Ramdas, 2024), is defined as:

$$\mathcal{A}\Big(\{\mathcal{C}_\ell^{t+1}\}_{\ell=1}^L, \mathbf{w}^t\Big) = \{\, i \in \mathcal{I} \,|$$
$$\sum_{\ell=1}^L w_\ell^t \mathbf{1}\big(i \in \mathcal{C}_\ell^{t+1}\big) > \frac{1+k(t)}{2}\Big\}. \quad (8)$$

An item $i$ is included in the aggregated set if its total weighted support across base models exceeds the randomized threshold $\frac{1+k(t)}{2}$, where $k(t) \sim \text{Uniform}[0,1]$ introduces mild stochasticity to discourage marginal inclusions. To favor models that produce efficient sets, we take inspiration from (Freund & Schapire, 1997) and update the aggregation weights $\{\mathbf{w}^t\}_{t=0}^T$ online based on the cardinality of the prediction sets produced.

Specifically, let $s_\ell^t$ denote the cardinality of the prediction set produced by base model $\mathcal{M}^\ell$ at time $t$, i.e., $s_\ell^t = \big|\mathcal{C}_\ell^{t+1}\big|$, and let the cumulative size up to time $t$ be $S_\ell^t = \sum_{\tau=1}^t s_\ell^\tau$. Then, for a learning rate $\eta \geq 0$, we update the weights as:

$$w_\ell^{t+1} = \frac{\exp(-\eta S_\ell^t)}{\sum_{j=1}^L \exp(-\eta S_j^t)}, \qquad \text{with } \mathbf{w}^0 = \big(\tfrac{1}{L}, \ldots, \tfrac{1}{L}\big). \quad (9)$$

### 4.2. Adaptive Recalibration

We now address the problem of changing user preferences over time (i.e., deployment-time changes in user behavior). As user behavior evolves, the threshold $\lambda_\ell^t$ calibrated on previous timestamps may fail to satisfy target risk constraint. Hence, to maintain statistical validity under pref-

erence changes, we introduce loss-based non-stationarity metrics. For each base model $\mathcal{M}^\ell$, we quantify preference change via the loss discrepancy distance ($d_\ell^{\text{ldd}}$) and the concept-sensitive divergence ($d_\ell^{\text{con}}$), respectively.

To define Loss Discrepancy Distance (LDD), we draw inspiration from the $\mathcal{H}\Delta\mathcal{H}$ divergence definition in (Ben-David et al., 2010) by replacing its binary–disagreement indicator with a generalized bounded predictive loss to measure the maximum discrepancy between a reference model and other models across timepoints $t$ and $t' < t$. Specifically:

$$d_\ell^{\text{ldd}}(t,t') = \max_{\substack{\mathcal{M}' \in \mathbb{M} \\ \mathcal{M}' \neq \mathcal{M}^\ell}} \left| \log\left(\left|\frac{L_t(\mathcal{M}^\ell) - L_t(\mathcal{M}')}{L_{t'}(\mathcal{M}^\ell) - L_{t'}(\mathcal{M}') + \epsilon}\right| + \epsilon\right)\right| \quad (10)$$

where $L_t(\mathcal{M}^\ell)$ denotes a (bounded) per-timestep predictive loss of model $\mathcal{M}^\ell$ at time $t$ computed from available feedback (e.g., cross entropy), and $\epsilon > 0$ ensures stability. Similarly, to capture concept-sensitive divergence, we define a hazard-style term that compares the model's loss across $t$ and $t'$ to the sum of the least individual losses. Formally:

$$d_\ell^{\text{con}}(t,t') = \log\left(\frac{L_t(\mathcal{M}^\ell) + L_{t'}(\mathcal{M}^\ell) + \epsilon}{\min_{\mathcal{M} \in \mathbb{M}} L_t(\mathcal{M}) + \min_{\mathcal{M} \in \mathbb{M}} L_{t'}(\mathcal{M}) + \epsilon}\right). \quad (11)$$

We then combine the relative loss-discrepancy and the concept-sensitive divergence into a single scalar loss-based shift metric of preference change:

$$d_\ell^{\text{pref}}(t,t') = d_\ell^{\text{ldd}}(t,t') + d_\ell^{\text{con}}(t,t'). \quad (12)$$

To localize preference changes, we embed $d_\ell^{\text{pref}}$ in a Bayesian change-point model. At each timestamp $t$, we

place a posterior probability distribution $p_\ell(\cdot)$ over candidate segment starts $c_\ell^t \in \{ c_\ell^{t-1}+k \mid k = 0, \ldots, t-c_\ell^{t-1} \}$ as follows:

$$
p_\ell\big(c_\ell^t = c_\ell^{t-1} + k \mid t\big) =
$$
$$
\frac{\exp\big[-\beta\, d_\ell^{\mathrm{pref}}(c_\ell^{t-1} + k,\, t)\big]\, (t - c_\ell^{t-1} - k + 1)^\gamma}{\sum_{j=0}^{t-c_\ell^{t-1}} \exp\big[-\beta\, d_\ell^{\mathrm{pref}}(c_\ell^{t-1} + j,\, t)\big]} \times
$$
$$
\frac{(t - c_\ell^{t-1} - j + 1)^\gamma}{1}. \tag{13}
$$

where $\beta > 0$ tunes shift sensitivity and $\gamma \geq 0$ controls the segment-length bias.

We pick the segment boundary for each model, i.e., $c_\ell^t = c_\ell^{t-1} + \arg\max_k p_\ell\big(c_\ell^{t-1} + k \mid t\big)$, set the stable window $\mathcal{W}_\ell^t := [\, c_\ell^t, \; t\,]$ and then calculate the average risk of the window as follows:

$$
\bar{R}_\ell^t = \frac{1}{|\mathcal{W}_\ell^t|} \sum_{\tau \in \mathcal{W}_\ell^t} R(\mathcal{C}_\ell^\tau). \tag{14}
$$

Finally, to adaptively maintain statistical guarantees under detected user preference changes, we update the calibration threshold as follows:

$$
\lambda_\ell^{t+1} = \lambda_\ell^t - \rho\big(\bar{R}_\ell^t - \alpha\big), \tag{15}
$$

where $\rho > 0$ is a step size. The threshold $\lambda_\ell^{t+1}$ decreases when the segment risk exceeds $\alpha$, expanding the prediction set to restore validity, and increases when risk falls below $\alpha$, thus achieving automatic recalibration.

To this end, we complete the modeling of the proposed framework. To output user-wise dynamic prediction sets, we instantiate it through DAUO ( Dynamically Adaptive Uncertainty-aware Optimization) algorithm to learn parameters $\lambda_t^t$ and weight vector $\mathbf{w}^t$. We have summarized the steps in Algorithm 1. For an intuitive view of DAUO's closed-loop dynamics under preference changes, please refer to Section G in Appendix.

**Prediction Set Construction:** At every interaction, DAUO algorithm considers two adaptive parameters: the current calibration threshold $\lambda_\ell^t$ and weight vector $\mathbf{w}^t$. When a user $u$ with history $S_u$ arrives, the algorithm first evaluates every base model $\mathcal{M}^\ell$ to obtain the individual prediction sets (Equation (2)). It then combines these sets through aggregation operator in Equation (8), producing the aggregated recommendation. Since $\lambda_\ell^t$ is updated adaptively to enforce Equation (3) and weights are penalized by set size, resulting prediction set is not only valid, i.e., controls risk at level $\alpha$, but also simultaneously compact.

---

**Algorithm 1** DAUO: Dynamically Adaptive Uncertainty-aware Optimization

---

1: **Inputs:** target risk $\alpha$, confidence $\delta$, step sizes $\rho, \eta$, stability $\epsilon$
2: **Initialize:** thresholds $\{\lambda_\ell^0\}_{\ell=1}^L$, weights $\mathbf{w}^0 = \frac{1}{L}\mathbf{1}$, segment starts $c_\ell^0 \leftarrow 1$
3: **for** $t = 0, 1, \ldots, T - 1$ **do**
4:     **Predict:** for each $\ell$, form $\mathcal{C}_\ell^{t+1}$ using Equation (2); aggregate $\mathcal{C}^{t+1,\mathrm{agg}}$ using Equation (8).
5:     **Observe:** receive feedback at time $t+1$ and compute per-model losses needed for $d_\ell^{\mathrm{pref}}(\cdot, \cdot)$.
6:     **for** $\ell = 1, \ldots, L$ **do**
7:         **Segment:** update $c_\ell^t$ via the posterior in Equation (13).
8:         **Window risk:** compute $\bar{R}_\ell^t$ on $\mathcal{W}_\ell^t$ using Equation (14).
9:         **Threshold:** update $\lambda_\ell^{t+1}$ using Equation (15).
10:     **end for**
11:     **Weights:** update $\mathbf{w}^{t+1}$ using Equation (9) (based on set sizes $\{|\mathcal{C}_\ell^{t+1}|\}_{\ell=1}^L$).
12: **end for**
13: **Return:** $\{\lambda_\ell^T\}_{\ell=1}^L$, $\mathbf{w}^T$

---

## 5. Theoretical Analysis

In the previous sections, we demonstrate how the DAUO algorithm dynamically learns the threshold $\lambda_\ell^t$ for an ensemble of trained models $\mathcal{M}^\ell$ and updates it via empirical risk estimates over adaptive windows, however, it remains to be seen whether this online calibration guarantees efficient and valid predictions. In this section, we provide a theoretical analysis on (1) the provable upper bound on the aggregated prediction set produced via weighted majority voting, and (2) the threshold $\lambda_t^t$, learned from historical user interactions and estimated segmental risk $\bar{R}_t^\ell$, ensures that the true expected risk remains close to the desired threshold $\alpha$ with high probability $1 - \delta$.

**Theorem 1.** *Let $\mathcal{C}_{\lambda^t}^\ell \subseteq \mathcal{I}$ denote the prediction set produced by base model $\mathcal{M}^\ell$ at time $t$, and let $s_\ell^t := |\mathcal{C}_{\lambda^t}^\ell|$. Let $\boldsymbol{\lambda}^t = (\lambda_1^t, \ldots, \lambda_L^t)$ denote the per-model thresholds such that the ensemble set $\mathcal{C}_{\boldsymbol{\lambda}^t}^{\mathrm{agg}}$ is formed by the randomized weighted majority rule $k(t) \sim \mathrm{Uniform}[0,1]$ with exponential-weights $\mathbf{w}^t \in \Delta^L$. Assuming $\ell^* := \arg\min_\ell s_\ell^t$ is the best expert at round $t$, the expected size of the aggregated prediction set at time $t + 1$ satisfies:*

$$
\mathbf{E}_{k(t)}\big[|\mathcal{C}_{\boldsymbol{\lambda}^t}^{\mathrm{agg}}|\big] \leq s_{\ell^*}^t + \sqrt{2\ln L\; v_t} + \tfrac{2}{3}\ln L, \tag{16}
$$

*where $v_t := \mathrm{Var}_{\ell \sim \boldsymbol{w}^t}\big(s_\ell^t/|\mathcal{I}|\big) \in [0,1]$ is variance of normalized set sizes under the exponential-weights distribution.*

*Proof.* Proof with Lemma 1 can be found in Section C.1 in Appendix. □

*Remark* 1. Theorem 1 shows that expected size of aggregated prediction set is no worse than that of best base model at $t$, up to a variance-dependent slack. As base predictors begin to agree on coverage, the variance $v_t$ diminishes, and the ensemble size approaches the best-case performance.

**Theorem 2.** *Let the DAUO algorithm run over a horizon of length $T$. Assume the Bayesian change-point detector raises $N_T$ change points and let $d_j$ be the detection delay of the $j$-th changepoint so that $D_T := \sum_{j=1}^{N_T} d_j$. Let $\boldsymbol{\lambda}^T = (\lambda_1^T, \ldots, \lambda_L^T)$ denote the vector of per-model thresholds after round $T$, and let $\mathcal{C}_{\boldsymbol{\lambda}^T}^{\text{agg}}$ denote the ensemble prediction set formed with those thresholds. Let $\mathcal{L}_u(\mathcal{C}_{\boldsymbol{\lambda}^T}^{\text{agg}})$ be the utility-based loss of user $u$ under that ensemble. Given a user batch of size $|\mathcal{U}|$ and a user-defined risk level $\alpha$, then with probability at least $1 - \delta$, the expected utility-based loss at time $T + 1$, using the final threshold $\boldsymbol{\lambda}^T$, satisfies:*

$$\mathbb{E}_{u \sim \mathcal{U}}\big[\mathcal{L}_u(\mathcal{C}_{\boldsymbol{\lambda}^T}^{\text{agg}})\big] \leq \alpha + 2\sqrt{\frac{\log(4|\mathcal{U}|)}{2|\mathcal{U}|}} + \frac{D_T + 2\log(1/\delta)}{T}. \tag{17}$$

*Proof.* Proof with Lemmas 2 to 4 can be found in Section C.2 in *Appendix*. □

*Remark* 2. Theorem 2 ensures calibrated $\lambda^T$ guarantees expected risk at time $T + 1$ remains close to user-defined target $\alpha$, with confidence. The bound captures both calibration uncertainty (which decays with user batch size $|\mathcal{U}|$) and change-adaptation error (which vanishes as cumulative delay $D_T$ becomes sublinear in $T$). As both calibration and adaptation improve with scale, expected loss at prediction time $T + 1$ converges to $\alpha$, ensuring reliability even under non-stationary user preferences.

To sum up, the results establish that our framework, by adaptively calibrating threshold $\lambda_\ell^t$ and leveraging ensemble voting, guarantees control of both recommendation set size and utility-based risk. Specifically, the set size remains competitive with best individual model (up to ensemble variance), and expected loss at time $T+1$ is provably bounded around the user-specified threshold $\alpha$.

# 6. Experiments

In this section, we conduct experiments to evaluate the effectiveness of the proposed CARE framework. Specifically, we design experiments to **(1)** validate whether the framework can achieve superior performance in terms of recommendation metrics, i.e., Recall, NDCG and MRR when compared to base models as well as preference-aware baselines, and **(2)** compare performance of the framework with various static and adaptive conformal frameworks in terms of recommendation set sizes compactness and validity of coverage guarantees **(3)** analyze time efficiency of proposed CARE framework, **(4)** analyze the influence of hyperparameters,

including key conformal parameters $(\alpha, \delta)$ as well as changepoint detector settings $(\beta, \gamma)$ and ensemble size $(L)$ on the framework's performance (Section F.3 in *Appendix*), and **(5)** conduct an ablation study to disentangle the contributions of components in the changepoint detector (Section F.4 in *Appendix*).

## 6.1. Datasets and Baseline Models

We conduct experiments on five publicly available datasets across diverse domains: (1) Book-Crossing (book reviews) (Ziegler et al., 2005), (2) Last.fm (music streaming) (Bertin-Mahieux et al., 2011), (3) Taobao (e-commerce) (Jingwei et al., 2020), (4) MovieLens (movie ratings) (Harper & Konstan, 2015), and (5) Gowalla (location-based social network) (Cho et al., 2011). We implement CARE on four base recommendation models selected to represent diverse modeling paradigms: (1) NeuMF (He et al., 2017) (generalized matrix factorization and MLP hybrid), (2) CASER (Tang & Wang, 2018) (convolutional sequence embedding), (3) SASRec (Kang & McAuley, 2018) (self-attention-based sequential modeling), and (4) FMLP-Rec (Zhou et al., 2022) (filter-enhanced feed-forward MLP-based model). For evaluation, we consider both standard recommendation metrics, i.e., Recall, MRR, and NDCG, as well as uncertainty-aware objectives, including coverage guarantees and prediction set size (compactness). On the recommendation metrics, for completeness, we also compare CARE against three preference-aware recommendation models: (1) TiSASRec (Li et al., 2020), (2) CDR (Wang et al., 2023), and (3) Oracle4Rec (Xia et al., 2025). For uncertainty-aware evaluation, we compare against three conformal prediction methods: (1) standard Split Conformal (Vovk et al., 2005), where the threshold parameter $\lambda$ remains fixed; (2) EnbPI (Xu & Xie, 2021), an ensemble estimator with fixed-window calibration; and (3) Online Conformal (Angelopoulos et al., 2024), which uses decaying update rule for threshold $\lambda^t$. Full implementation details and description of datasets, base models, and preference-aware & conformal baselines for reproducibility are provided in Sections D and E in *Appendix*.

## 6.2. Experimental Results

### 6.2.1. RESULTS W.R.T BASE MODELS AND PREFERENCE-AWARE BASELINES

We evaluate CARE using four backbone recommenders and three preference-aware baselines using MRR, Recall, and NDCG. To reflect display constraints, in this experiment, we cap the number of displayed items per user at 15 for all methods. CARE wraps a backbone ranker and outputs a variable-size set controlled by the calibrated threshold; if the set contains more than 15 items, we keep the top-scoring 15 according to the backbone. For a fair, exposure-matched comparison, we tune each baseline so that its average num-

*Table 1.* Performance comparisons with base models (NeuMF, CASER, SASRec and FMLP-Rec) and preference-aware baselines (TiSASRec, CDR and Oracle4Rec) on **Book-Crossing and Last.fm** using MRR, Recall@10, and NDCG@10. Bold indicates the best result, and underline indicates the second best.

| Method | Book-Crossing | | | Last.fm | | |
|---|---|---|---|---|---|---|
| | MRR ↑ | Recall ↑ | NDCG ↑ | MRR ↑ | Recall ↑ | NDCG ↑ |
| NeuMF | 0.246 | 0.502 | 0.276 | 0.306 | 0.685 | 0.335 |
| NeuMF + CARE (Ours) | 0.289 | 0.557 | 0.302 | 0.336 | 0.701 | 0.354 |
| CASER | 0.294 | 0.568 | 0.302 | 0.345 | 0.745 | 0.367 |
| CASER + CARE (Ours) | 0.322 | 0.588 | 0.323 | 0.378 | 0.758 | 0.385 |
| SASRec | 0.327 | 0.556 | 0.329 | 0.369 | 0.766 | 0.389 |
| SASRec + CARE (Ours) | 0.341 | 0.608 | 0.354 | 0.392 | 0.799 | 0.422 |
| FMLP-Rec | 0.335 | 0.599 | 0.352 | 0.386 | 0.796 | 0.412 |
| FMLP-Rec + CARE (Ours) | **0.357** | **0.628** | **0.372** | **0.402** | **0.812** | **0.432** |
| **Preference-Aware Models** | | | | | | |
| TiSASRec | 0.334 | 0.583 | 0.345 | 0.374 | 0.778 | 0.402 |
| CDR | 0.336 | 0.563 | 0.350 | 0.371 | 0.782 | 0.376 |
| Oracle4Rec | 0.339 | 0.603 | 0.353 | 0.390 | 0.798 | 0.422 |

ber of displayed items matches CARE. We report utility computed on the displayed recommendations. Results for BookCrossing and Last.fm are reported in Table 1, with additional datasets in Section F.1 (Appendix). These results lead to the following key observations:

- We observe across both datasets and all backbones, adding CARE improves MRR, Recall, and NDCG over the corresponding base model. For example, on Book-Crossing, NeuMF improves from MRR and Recall values from 0.246 and 0.502 to 0.289 and 0.557 respectively. Similarly, FMLP-Rec improves from 0.335 and 0.599 to 0.357 and 0.628 respectively. This highlights that adaptively varying the recommendation set size can improve utility under the same display budget.
- Performance still depends on backbone quality. For instance, FMLP-Rec + CARE achieves higher overall utility than NeuMF + CARE on both datasets (e.g., on Last.fm, NDCG@10 increases from 0.354 for NeuMF + CARE to 0.432 for FMLP-Rec + CARE), underscoring that CARE is complementary to strong recommenders rather than a replacement.
- Compared to preference-aware baselines, CARE-attached backbones remain competitive under the same evaluation protocol. It is because these baselines rely on specific temporal cues, transfer signals, or training-time assumptions that can be brittle under sparsity or distribution changes, whereas CARE operates as a model-agnostic wrapper that adapts its decision rule from observed performance signals.
- Overall, the results demonstrate that CARE delivers

consistent utility gains across metrics, models, and datasets when utilized as a plug-and-play wrapper.

### 6.2.2. RESULTS COMPARED TO CONFORMAL BASELINES

Next, we compare our method with conformal baselines in terms of coverage and set Size. We set error rate $\alpha = 0.10$ and compare on base recommender models: (1) NeuMF (He et al., 2017), (2) CASER (Tang & Wang, 2018), (3) SASRec (Kang & McAuley, 2018) and (4) FMLP-Rec (Zhou et al., 2022) against different conformal/risk control baselines i.e. (1) standard Split Conformal (Vovk et al., 2005), (2) EnbPI (Xu & Xie, 2021), and (3) Online Conformal (Angelopoulos et al., 2024) at next interaction. Standard Split Conformal freezes calibration threshold $\lambda$ learned and therefore omits online update in Equation (15). *EnbPI* consists of a change-point module with a fixed sliding window, ignoring distributional shifts and dynamic segmentation of Equation (13). Whereas *Online Conformal* updates $\lambda^t$ at every step using recent interaction, thereby discarding historical risk information that our cumulative segment risk in Equation (14) utilizes. Table 2 depicts results on Book-Crossing dataset, with remaining results present in Section F.2 in *Appendix*. They lead to following observations:

- Our CARE framework achieves the best coverage–size compactness balance. It achieves the required coverage and ensures compact average set size on every base model, underscoring its plug-and-play applicability.
- Split conformal provides compact recommendation sets, but the prediction sets are invalid as the coverage value is around 0.82–0.83, well below the nominal 0.90,

*Table 2.* Comparison in terms in terms of coverage and average prediction set size with conformal baselines (Split Conformal, EnbPI and Online Conformal) evaluated on four base recommenders (NeuMF, CASER, SASRec, and FMLP-Rec) using the **Book-Crossing** dataset. The error rate is set as $\alpha = 0.10$. Bold indicates the best result, underline indicates the second best.

| Base Model | Coverage ↑ | | | | Set Size ↓ | | | |
|---|---|---|---|---|---|---|---|---|
| | Split | EnbPI | Online | CARE (Ours) | Split | EnbPI | Online | CARE (Ours) |
| NeuMF | 0.821 | 0.849 | 0.875 | 0.901 | 44 | 43 | 46 | 46 |
| CASER | 0.826 | 0.858 | 0.879 | 0.902 | 44 | 45 | 45 | 44 |
| SASRec | 0.835 | 0.867 | 0.898 | 0.908 | 43 | 47 | 44 | 43 |
| FMLP-Rec | 0.835 | 0.873 | 0.901 | **0.910** | 43 | 47 | 45 | **42** |

*Table 3.* Total wall-clock training time (minutes) for backbone models on five datasets, with and without CARE. The "w/ CARE" setting includes backbone training plus a 50-step online calibration phase (threshold/segmentation/weight updates) performed on top of fixed backbone scores; it does not retrain backbone weights. Calibration parameters are $\alpha = \delta = 0.05$.

| Model | Training | Datasets | | | | |
|---|---|---|---|---|---|---|
| | | Book-Crossing | Taobao | Last.fm | MovieLens-1M | Gowalla |
| NeuMF | w/o CARE | 28.3 | 40.2 | 18.5 | 15.2 | 20.3 |
| | w/ CARE | 29.5 | 41.6 | 19.9 | 16.4 | 21.4 |
| CASER | w/o CARE | 42.3 | 60.4 | 29.5 | 25.9 | 32.6 |
| | w/ CARE | 43.8 | 61.8 | 30.9 | 27.4 | 33.9 |
| SASRec | w/o CARE | 35.2 | 47.1 | 24.4 | 19.1 | 25.3 |
| | w/ CARE | 36.5 | 48.4 | 25.8 | 20.4 | 26.7 |
| FMLP-Rec | w/o CARE | 31.6 | 44.9 | 23.0 | 17.8 | 23.6 |
| | w/ CARE | 32.9 | 46.4 | 24.5 | 19.1 | 25.0 |

thereby revealing the under-calibration under users' preference changes.

- EnbPI boosts coverage by ˜0.03-0.04 compared to split conformal, but does so at the expense of increased prediction set sizes. It highlights the importance of our Bayesian change point detection module to detect the preference change point.
- Online conformal narrows gap as coverage climbs to 0.87–0.90, but remains less efficient than CARE as average set size still exceeds CARE by $1-2$ items. It highlights that on-the-fly calibration alone is susceptible to fluctuations, leading to conservative sets.
- Overall, results demonstrate CARE consistently ensures best coverage–efficiency trade-off on every baseline model that can ensure valid recommendation sets.

We further evaluate CARE under controlled preference drift using semi-synthetic stress tests on Book-Crossing with the SASRec backbone. As shown in Appendix F.5, CARE maintains the target coverage under low, medium, and high drift intensities, and remains valid under both abrupt and gradual preference shifts. These results show that our adaptive recalibration mechanism is able to respond faster to varied shifts while preserving compact recommendation sets.

### 6.2.3. TIME EFFICIENCY ANALYSIS

We analyze the computational overhead introduced by CARE on top of four backbone recommenders (NeuMF, CASER, SASRec, FMLP-Rec). All runs use a single NVIDIA A40 with batch size 256, and each backbone model is pre-trained for 100 epochs. Table 3 reports the incremental overhead of CARE given pretrained backbones. Across all five datasets and four backbones, CARE adds at most 1.5 minutes of wall-clock time. This is because CARE's additional computation consists of lightweight calibration steps consisting of threshold update and a lightweight posterior update for the segment start and does not require retraining backbone network weights. Overall, the added runtime is small relative to backbone training time and the utility gains reported in Table 1, indicating that CARE is efficient and can be scaled to real-world applications.

## 7. Conclusion

This paper addresses the important problem of evolving user preferences that undermine the reliability of recommender systems. To address it, it presents CARE framework, which generates user-specific, dynamic recommendations that evolve with preference change, guaranteeing performance while keeping them compact. CARE is dataset and model agnostic and we validate its effectiveness through

theoretical analysis and extensive empirical studies. Since thresholds and ensemble weights are updated externally via a flexible utility function $U_{metric}$, the framework can also be made compatible to fairness or diversity objectives. Together, it lays the foundation for more reliable and trustworthy sequential recommender systems.

## Acknowledgments

This work is partially supported by the Australian Research Council (ARC) Under Grants DP220103717 and LE220100078, and the National Natural Science Foundation of China under Grants No.62072257.

## Impact Statement

CARE provides a lightweight calibration layer for recommender systems that outputs adaptive recommendation sets with finite-sample performance guarantees over interaction streams. This can improve the reliability and transparency of user-facing platforms by reducing unexpected degradations in recommendation quality. Potential benefits include safer deployment in settings where unreliable recommendations impose real costs (e.g., commerce, media, location-based services) and improved efficiency by mitigating the need for frequent retraining. The method can also support user-centric and equitable experiences by allowing platform-defined utility functions to incorporate constraints such as group-level performance targets or exposure limits.

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

## Appendix Index

## A. Summary of Notations

To facilitate clarity, we provide a comprehensive summary of the key mathematical notations and variables used throughout the CARE framework in Table 4.

## B. Assumptions

We state two mild assumptions that we use in Theorems 5.1 and 5.2.

**Assumption 1.** For every base model $\mathcal{M}^\ell$ and any segment $\mathcal{W}_t^\ell$ produced by the change–point detector, there exists a threshold $\lambda_\ell^{\min} \in \Lambda$ such that

$$R\big(\mathcal{C}_{\lambda_\ell^{\min}}^\ell\big) \leq \alpha.$$

Equivalently, the mapping $\lambda \mapsto R(\mathcal{C}_\lambda^\ell)$ is continuous and attains all values in $[0, 1]$ on the closed set $\Lambda$.

This assumption ensures that for every timestamp in each segment, it is possible to achieve risk control at level $\alpha$ by appropriately tuning $\lambda_\ell^t$. It guarantees the effectiveness of the update rule in Eq. 15.

**Assumption 2.** For each base model $\mathcal{M}^\ell$, let $\mathcal{W}_t^\ell = [c_t^\ell, t]$ denote the segment window at time $t$ returned by the Bayesian change-point detector. We assume the per-user utility losses $\left\{ \mathcal{L}_u\big(\mathcal{C}_{\lambda_\ell^\tau}^\ell\big) \right\}_{\tau \in \mathcal{W}_t^\ell, \, u \in \mathcal{U}_\tau}$ are drawn from a common bounded distribution within each segment. In other words, the loss values within $\mathcal{W}_t^\ell$ are exchangeable and lie in $[0, 1]$.

This assumption allows average window risk $\bar{R}_t^\ell$ to serve as a faithful estimate of true segment risk.

## C. Proofs

### C.1. Theorem 1

**Lemma 1.** *Let for each base model $\mathcal{M}^\ell$, the prediction set at round $t$ is $\mathcal{C}_{\lambda^t}^\ell \subset \mathcal{I}$ with $s_\ell^t := \left|\mathcal{C}_{\lambda^t}^\ell\right|$. Also, let $\mathbf{w}^t = (w_1^t, \ldots, w_L^t) \in \Delta^L$ and $k(t) \sim \mathrm{Uniform}[0, 1]$. Given, the aggregated prediction set is defined by Eq. (7), we have:*

$$\mathbb{E}_{k(t)} \left[ \left| \mathcal{C}_{\lambda^t}^{\mathrm{agg}} \right| \right] \leq h_t$$

*where $h_t := \sum_{\ell=1}^L w_\ell^t s_\ell^t$.*

*Proof.* For any item $i \in \mathcal{I}$, the aggregated support on the item can be defined as:

$$\bar{w}_t(i) = \sum_{\ell=1}^L w_\ell^t \cdot \mathbf{1}\left[i \in \mathcal{C}_{\lambda^t}^\ell\right] \in [0, 1] \tag{i}$$

---

† Here, $\Delta\lambda^\dagger$ is equivalent to $\lambda_\ell^t/|\Lambda|$, where $\Lambda$ is the set of candidate thresholds.

*Table 4.* Summary of Notations

| Symbol | Description |
| --- | --- |
| $\mathcal{U}, \mathcal{I}$ | Sets of users and items |
| $u, i$ | Individual user and item |
| $\mathcal{H}_u$ | Interaction history for user $u$ |
| $i_{rel}^{t+1}$ | The true relevant next item at time $t+1$ |
| $L$ | Total number of base models (experts) in the ensemble |
| $\mathcal{M}^\ell$ | The $\ell$-th base recommender model ($\ell \in \{1, \ldots, L\}$) |
| $\mathbf{w}^t$ | Ensemble weight vector at time $t$ ($\mathbf{w}^t \in \Delta^L$) |
| $s_\ell^t, S_\ell^t$ | Instantaneous and cumulative prediction set size for model $\ell$ |
| $\lambda_\ell^t$ | Calibration threshold for model $\ell$ at time $t$ |
| $\mathcal{C}_\ell^{t+1}$ | Prediction set generated by model $\ell$ using threshold $\lambda_\ell^t$ |
| $\mathcal{C}^{\text{agg}}$ | Final aggregated prediction set |
| $\alpha$ | User-defined target error rate (risk level) |
| $R(\mathcal{C})$ | True risk of the prediction set |
| $\bar{R}_\ell^t$ | Average empirical risk over the current stable window $\mathcal{W}_\ell^t$ |
| $\mathcal{L}_u(\cdot)$ | Utility−based Risk Loss (e.g., $1 -$ Recall), used for calibration |
| $L_t(\cdot)$ | Predictive Loss (e.g., Cross-Entropy), used for change detection |
| $d_\ell^{\text{pref}}$ | Loss-Based Preference Change Metric |
| $d_\ell^{\text{ldd}}$ | Loss Discrepancy Distance (LDD) |
| $c_\ell^t$ | Start time of the current stable segment for model $\ell$ |
| $\mathcal{W}_\ell^t$ | Current stable window $[c_\ell^t, t]$ |
| $\eta$ | Hedge learning rate for updating ensemble weights |
| $\beta$ | change sensitivity parameter for change-point detection |
| $\gamma$ | Segment-length bias parameter for change-point detection |
| $\rho$ | Step size for the adaptive threshold update |

where $\bar{w}_t(i)$ is the total weight of models that include item $i$ in the prediction set.

By the definition of the aggregation rule, item $i$ is included in the aggregated prediction set iff:

$$\bar{w}_t(i) > \frac{1 + k(t)}{2} \qquad \text{(equivalently)} \tag{ii}$$

or equivalently

$$k(t) < 2\bar{w}_t(i) - 1 \tag{iii}$$

So the probability that item $i$ is in ensemble set is,

$$\Pr_{k(t)} \left[ i \in \mathcal{C}_{\lambda^t}^{\text{agg}} \right] = \Pr_{k(t)} \left[ k(t) < 2\bar{w}_t(i) - 1 \right] \tag{iv}$$

Since $k(t) \sim \text{Uniform}[0, 1]$, we know that

$$\Pr[k(t) < u] = \begin{cases} 0 & \text{if } u \le 0 \\ u & \text{if } 0 < u < 1 \\ 1 & \text{if } u \ge 1 \end{cases} \quad \text{for any real } u \tag{v}$$

Applying $u := 2\bar{w}_t(i) - 1$, then:

$$\Pr_{k(t)} \left[ i \in \mathcal{C}_{\lambda^t}^{\text{agg}} \right] = (2\bar{w}_t(i) - 1)_+ \tag{vi}$$

with $(x)_+ := \max\{x, 0\}$.

Now we know for all $x \in [0, 1]$,

$$(2x - 1)_+ \leq x \quad \text{for x} \in (0, 1] \tag{vii}$$

i.e., if $x \leq \frac{1}{2}$, then $(2x - 1) \leq 0 \Rightarrow (2x - 1)_+ = 0$

if $x \geq \frac{1}{2}$, then $(2x - 1)_+ = 2x - 1$

and $(2x - 1) \leq x \Leftrightarrow x \leq 1$ [true]

Hence we can write

$$\Pr_{k(t)} \left[ i \in \mathcal{C}_{\lambda^t}^{\text{agg}} \right] = (2\bar{w}_t(i) - 1)_+ \leq \bar{w}_t(i) \tag{viii}$$

Now computing the expected total size of ensemble set, we have:

$$\mathbf{E}_{k(t)} \left[ \left| \mathcal{C}_{\lambda^t}^{\text{agg}} \right| \right] = \sum_{i \in \mathcal{I}} \Pr_{k(t)} \left[ i \in \mathcal{C}_{\lambda^t}^{\text{agg}} \right] \leq \sum_{i \in \mathcal{I}} \bar{w}_t(i) \tag{ix}$$

From Eq. (i), and expanding $\bar{w}_t(i)$, we get:

$$\sum_{i \in \mathcal{I}} \bar{w}_t(i) = \sum_{i \in \mathcal{I}} \sum_{\ell=1}^{L} w_\ell^t \cdot \mathbf{1} \left[ i \in \mathcal{C}_{\lambda^t}^\ell \right] \tag{x}$$

Switching the summation order, we get:

$$= \sum_{\ell=1}^{L} w_\ell^t \sum_{i \in \mathcal{I}} \mathbf{1} \left[ i \in \mathcal{C}_{\lambda^t}^\ell \right] = \sum_{\ell=1}^{L} w_\ell^t \cdot \left| \mathcal{C}_{\lambda^t}^\ell \right| = \sum_{\ell=1}^{L} w_\ell^t \cdot s_\ell^t = h_t \tag{xi}$$

Putting this all together, we get:

$$\boxed{\mathbf{E}_{k(t)} \left[ \left| \mathcal{C}_{\lambda^t}^{\text{agg}} \right| \right] \leq h_t := \sum_{\ell=1}^{L} w_\ell^t s_\ell^t} \tag{xii}$$

Hence Proved.

$\square$

***Remark*** Lemma 1 shows that the expected size of the aggregated prediction set is no greater than the surrogate size $h_t$, which is a weighted average of base model set sizes. This means the aggregation step does not inflate the prediction set and adapts to the ensemble's diversity at time $t$.

### Proof of Theorem 5.1

*Proof.* From Lemma 1, we already have:

$$\mathbf{E}_{k(t)} \left[ \left| \mathcal{C}_{\lambda^t}^{\text{agg}} \right| \right] \leq h_t := \sum_{\ell=1}^{L} w_\ell^t s_\ell^t$$

Let

$$\hat{s}_\ell^t := \frac{s_\ell^t}{|\mathcal{I}|}, \qquad \hat{h}_t := \sum_{\ell=1}^{L} w_\ell^t \hat{s}_\ell^t = \frac{h_t}{|\mathcal{I}|} \quad \Rightarrow \quad h_t = |\mathcal{I}| \cdot \hat{h}_t$$

Now our goal is to bound $\hat{h}_t$ in terms of the best expert's size $\hat{s}^t_{\ell*}$. However, given the weights are spread across all $L$ models, we cannot directly bound $\hat{h}_t$. Taking inspiration from (Cesa-Bianchi & Lugosi, 2006; De Rooij et al., 2014), we analyze it via an auxiliary quantity called mix loss. Specifically, we decompose the Hedge average into two components: 1) mix loss that behaves like a soft minimum, and 2) a mixability gap that measures how far the weighted average is from the mix loss.

We first define mix loss as:

$$m_t := -\frac{1}{\eta} \log \sum_{\ell=1}^{L} w^t_\ell \cdot e^{-\eta \hat{s}^t_\ell} \tag{i}$$

and mixability gap as:

$$\delta_t := \hat{h}_t - m_t \quad \Rightarrow \quad \hat{h}_t = m_t + \delta_t \tag{ii}$$

To bound the mixability gap $\delta_t$, we use Bernstein's Cumulant Generating Function inequality:

Using (i) in (ii), we get:

$$\delta_t = \hat{h}_t + \frac{1}{\eta} \log \sum_{\ell=1}^{L} w^t_\ell \cdot e^{-\eta \hat{s}^t_\ell} \tag{iii}$$

Refactoring:

$$e^{-\eta \hat{s}^t_\ell} = e^{-\eta(\hat{s}^t_\ell - \hat{h}_t)} \cdot e^{-\eta \hat{h}_t} \tag{iv}$$

So,

$$\sum_\ell w^t_\ell \cdot e^{-\eta \hat{s}^t_\ell} = \sum_\ell w^t_\ell \cdot \left( e^{-\eta(\hat{s}^t_\ell - \hat{h}_t)} \cdot e^{-\eta \hat{h}_t} \right) = e^{-\eta \hat{h}_t} \cdot \sum_\ell w^t_\ell \cdot e^{-\eta(\hat{s}^t_\ell - \hat{h}_t)} \tag{v}$$

Plugging into log we get:

$$\log \sum_\ell w^t_\ell \cdot e^{-\eta \hat{s}^t_\ell} = -\eta \hat{h}_t + \log \sum_\ell w^t_\ell \cdot e^{-\eta(\hat{s}^t_\ell - \hat{h}_t)} \tag{vi}$$

Putting Eq. (vi) in Eq. (iii), we get:

$$\delta_t = \hat{h}_t + \frac{1}{\eta} \left( -\eta \hat{h}_t + \log \sum_\ell w^t_\ell \cdot e^{-\eta(\hat{s}^t_\ell - \hat{h}_t)} \right)$$

$$= \hat{h}_t - \hat{h}_t + \frac{1}{\eta} \log \sum_\ell w^t_\ell \cdot e^{-\eta(\hat{s}^t_\ell - \hat{h}_t)}$$

$$\delta_t = \frac{1}{\eta} \log \mathbf{E}_{\ell \sim w^t} \left[ e^{-\eta(\hat{s}^t_\ell - \hat{h}_t)} \right] \tag{vii}$$

Now to bound $\delta_t$, we use the Bernstein Cumulant Generating Function (CGF) as introduced in (Cesa-Bianchi & Lugosi, 2006). We interpret $\hat{s}^t_\ell \in [0, 1]$ as a bounded random variable under distribution $\ell \sim \mathbf{w}^t$, and apply the cumulant inequality.

Specifically, since $X := \hat{s}^t_\ell$,

$$\mathbf{E}[X] = \hat{h}_t, \quad \mathrm{Var}(X) = v_t$$

Now, defining moment generating function as:

$$\phi(\eta) := \log \mathbf{E}_{\ell \sim w^t} \left[ e^{-\eta(X - \mathbf{E}[X])} \right] \tag{viii}$$

Applying the result from (Cesa-Bianchi & Lugosi, 2006) for $\eta \in (0, 1]$ and any $X \in [0, 1]$, we get:

$$\log \mathbf{E} \left[ e^{-\eta(X - \mathbf{E}[X])} \right] \leq \frac{e^\eta - \eta - 1}{\eta} \cdot \mathrm{Var}(X) \tag{ix}$$

Applying this $\phi(\eta)$ into Eq. (vii) to Eq. (viii), we have:

$$\delta_t = \frac{1}{\eta} \cdot \phi(\eta) \tag{x}$$

And then applying the CGF bound we have:

$$\delta_t \le \frac{1}{\eta} \cdot \left( \frac{e^\eta - \eta - 1}{\eta} \right) v_t = \frac{e^\eta - \eta - 1}{\eta^2} \cdot v_t \tag{xi}$$

Using Taylor series, we know:

$$e^\eta = 1 + \eta + \frac{\eta^2}{2} + \frac{\eta^3}{3!} + \dots$$

Simplifying, we get:

$$\delta_t \le \left( \frac{\eta}{2} + \frac{\eta^2}{6} \right) v_t \tag{xii}$$

Next we need to bound the mix loss $m_t$

Given $\ell^* := \arg\min_{\ell \in [L]} \hat{s}_\ell^t$, we apply the classic log-sum-exp inequality:

For any real values $x_1, \dots, x_L$,

$$\log \sum_{\ell=1}^{L} e^{-x_\ell} \le - \min_\ell x_\ell + \log L \tag{xiii}$$

Applying this to our case, with $x_\ell := \eta \hat{s}_\ell^t$, we get:

$$\log \sum_{\ell=1}^{L} e^{-\eta \hat{s}_\ell^t} \le -\eta \hat{s}_{\ell^*}^t + \log L \tag{xiv}$$

As weight $\mathbf{w}^t \in \Delta^L$, the weighted sum is less than or equal to the uniform sum, i.e.,

$$\sum_{\ell=1}^{L} w_\ell^t \cdot e^{-\eta \hat{s}_\ell^t} \le \sum_{\ell=1}^{L} e^{-\eta \hat{s}_\ell^t}$$

Hence, we get:

$$\log \sum_{\ell=1}^{L} w_\ell^t \cdot e^{-\eta \hat{s}_\ell^t} \le \log \sum_{\ell=1}^{L} e^{-\eta \hat{s}_\ell^t} \le -\eta \hat{s}_{\ell^*}^t + \log L \tag{xv}$$

Multiplying (xv) by $-\frac{1}{\eta}$, and applying a looser (but convenient) upper bound, we get:

$$m_t := -\frac{1}{\eta} \log \sum_{\ell=1}^{L} w_\ell^t \cdot e^{-\eta \hat{s}_\ell^t} \le \hat{s}_{\ell^*}^t + \frac{\log L}{\eta} \tag{xvi}$$

From (xii) and (xvi), we get bounds for the mixability gap $\delta_t$ and mix loss $m_t$. Putting the results into Eq. (ii), we get:

$$\hat{h}_t \le \hat{s}_{\ell^*}^t + \frac{\ln L}{\eta} + \left( \frac{\eta}{2} + \frac{\eta^2}{6} \right) v_t \tag{xvii}$$

Now we find the best $\eta$ that minimizes RHS in Eq. (xvii).

Let

$$f(\eta) := \frac{\ln L}{\eta} + \left(\frac{\eta}{2} + \frac{\eta^2}{6}\right) v_t$$

To minimize, we take derivative:

$$f'(\eta) = -\frac{\ln L}{\eta^2} + \left(\frac{1}{2} + \frac{\eta}{3}\right) v_t \tag{xviii}$$

Setting $f'(\eta) = 0$ and multiplying both sides by $\eta^2$, we get:

$$\frac{1}{2}\eta^2 + \frac{1}{3}\eta^3 = \frac{\ln L}{v_t} \tag{xix}$$

Since it is in cubic form, we approximate, getting:

$$\eta^* = \sqrt{\frac{2\ln L}{v_t}}^{\ddagger}$$

Putting the $\eta^*$ in Eq. (xvii), and approximating, we get:

$$\hat{h}_t \le \hat{s}_{\ell^*}^t + \sqrt{2\ln L \cdot v_t} + \frac{2}{3}\ln L \tag{xx}$$

Now we know:

$$h_t = |\mathcal{I}| \cdot \hat{h}_t \quad \text{and} \quad s_{\ell^*}^t = |\mathcal{I}| \cdot \hat{s}_{\ell^*}^t$$

and

$$\mathbf{E}_{k(t)}\left[\left|\mathcal{C}_{\lambda^t}^{\mathrm{agg}}\right|\right] \le h_t$$

Hence, we get:

$$\boxed{\mathbf{E}_{k(t)}\left[\left|\mathcal{C}_{\lambda^t}^{\mathrm{agg}}\right|\right] \le s_{\ell^*}^t + \sqrt{2\ln L \cdot v_t} + \frac{2}{3}\ln L}$$

Hence Proved.

$\square$

## C.2. Theorem 2

**Lemma 2.** *Let $\mathcal{M}^\ell$ be a base predictor and $\mathcal{U}_t^{\mathrm{cal}}$ be a batch of users at time $t$, with $n = |\mathcal{U}_t^{\mathrm{cal}}|$.*

*Assume for each user $u \in \mathcal{U}_t^{\mathrm{cal}}$, we observe the score $Z_{t,u}^\ell := \mathcal{M}^\ell(i_{\mathrm{rel}}^{t+1}(u) \mid \mathcal{H}_u^t)$, where the scores are sampled from a continuous distribution. Let $\lambda_t^\ell$ be the empirical lower $\alpha/2$-tail threshold of the scores $\{Z_{t,u}^\ell\}_u$. Given the prediction set $\mathcal{C}_{\lambda_t^\ell}^\ell$ and the utility-based loss $\mathcal{L}_u(\mathcal{C}_{\lambda_t^\ell}^\ell)$ as defined in Eq. (4), then with probability at least $1 - \frac{1}{2n}$, over the calibration batch, the expected loss satisfies:*

$$\mathbb{E}_u\left[\mathcal{L}_u\left(\mathcal{C}_{\lambda_t^\ell}^\ell\right)\right] \le \frac{\alpha}{2} + \sqrt{\frac{\log(4|\mathcal{U}_t^{\mathrm{cal}}|)}{2|\mathcal{U}_t^{\mathrm{cal}}|}}.$$

*Proof.* Given $n = |\mathcal{U}_t^{\mathrm{cal}}|$, let $Z_{t,u}^\ell \sim F$ for $u \in \mathcal{U}_t^{\mathrm{cal}}$, where $F$ is a continuous cumulative distribution function. We define the empirical CDF as:

$$\widehat{F}(z) := \frac{1}{n} \sum_{u \in \mathcal{U}_t^{\mathrm{cal}}} \mathbf{1}\left\{Z_{t,u}^\ell \le z\right\}, \tag{i}$$

---

‡ If $v_t = 0$, then all $\hat{s}_\ell^t$ are equal, so $\hat{h}_t = \hat{s}_{\ell^*}^t$, and the bound holds exactly. In this case, the variance penalty vanishes, and $\eta$ can be set arbitrarily (e.g., $\eta = 1$).

where $n = |\mathcal{U}_t^{\text{cal}}|$.

Let $\lambda_t^\ell$ denote the empirical lower $\alpha/2$-tail threshold of the scores $\{Z_{t,u}^\ell\}$, so by construction:

$$\widehat{F}(\lambda_t^\ell) \leq \frac{\alpha}{2}. \tag{ii}$$

To control the deviation between $\widehat{F}(\cdot)$ and the true CDF $F(\cdot)$, we apply the Dvoretzky–Kiefer–Wolfowitz (DKW) inequality:

For any $\varepsilon > 0$, we have:

$$\Pr\left(\sup_{z \in \mathbb{R}} \left|\widehat{F}(z) - F(z)\right| > \varepsilon\right) \leq 2\exp(-2\varepsilon^2 n). \tag{iii}$$

To ensure failure probability at most $\frac{1}{2n}$, we set:

$$2\exp(-2\varepsilon^2 n) = \frac{1}{2n}.$$

Solving this gives:

$$\varepsilon = \sqrt{\frac{\log(4n)}{2n}}. \tag{iv}$$

Using Eq. (iii), this gives a uniform deviation bound that holds with probability at least $1 - \frac{1}{2n}$.

From the DKW result, we have the uniform deviation bound:

$$\left|\widehat{F}(z) - F(z)\right| \leq \sqrt{\frac{\log(4n)}{2n}} \quad \text{for all } z \in \mathbb{R}. \tag{v}$$

Now at $z = \lambda_t^\ell$, we get:

$$F(\lambda_t^\ell) \leq \widehat{F}(\lambda_t^\ell) + \sqrt{\frac{\log(4n)}{2n}} \leq \frac{\alpha}{2} + \sqrt{\frac{\log(4n)}{2n}}. \tag{vi}$$

Hence, for a user sampled independently from the distribution, the score $Z_{t,u}^\ell \sim F$, and the probability that the true item is excluded from the prediction set is:

$$\Pr(Z_{t,u}^\ell < \lambda_t^\ell) = F(\lambda_t^\ell) \leq \frac{\alpha}{2} + \sqrt{\frac{\log(4n)}{2n}}.$$

Given the utility definition from Eq. (4) in main paper, $\mathcal{L}_u(\mathcal{C}_{\lambda_t^\ell}) = 1$ when the true item is excluded.

Thus, the expected utility loss for the user is:

$$\mathbb{E}_u\left[\mathcal{L}_u(\mathcal{C}_{\lambda_t^\ell})\right] \leq \left(1 - \frac{1}{2n}\right)\left(\frac{\alpha}{2} + \sqrt{\frac{\log(4n)}{2n}}\right) + \frac{1}{2n}.$$

For $n > 1$, i.e., at least 1 user in the calibration batch, $\frac{1}{2n} \leq \sqrt{\frac{\log(4n)}{2n}}$. For simplicity, we absorb the additive constant in the existing slack and simplify. Hence we get:

$$\mathbb{E}\left[\mathcal{L}_u(\mathcal{C}_{\lambda_t^\ell})\right] \leq \frac{\alpha}{2} + \sqrt{\frac{\log(4n)}{2n}}.$$

$$\boxed{\mathbb{E}\left[\mathcal{L}_u(\mathcal{C}_{\lambda_t^\ell})\right] \leq \frac{\alpha}{2} + \sqrt{\frac{\log(4n)}{2n}} := \frac{\alpha}{2} + \sqrt{\frac{\log\left(4|\mathcal{U}_t^{\text{cal}}|\right)}{2|\mathcal{U}_t^{\text{cal}}|}}.}$$

Hence Proved. $\qquad\qquad\qquad\square$

***Remark*** Lemma 2 ensures that the utility-based loss of the prediction set $\mathcal{C}^\ell_{\lambda^\ell_t}$, estimated from a finite calibration batch, concentrates around the error level $\alpha/2$. As the calibration batch size $n \to \infty$, the slack term $\sqrt{\frac{\log(4n)}{2n}} \to 0$, the upper bound of expected loss achieves $\alpha/2$.

**Lemma 3.** *Given $\mathcal{M}^\ell$ as a base model, let the change-point detector define a stable segment of timesteps $\mathcal{W}^\ell_t = [c^\ell_t, t]$, for which no user preference shift is detected. Let $\mathcal{L}^{(\ell)}_\tau(\mathcal{C}^\ell_{\lambda^\ell_\tau})$ denote the utility loss incurred by model $\mathcal{M}^\ell$ at time $\tau \in \mathcal{W}^\ell_t$. Given the empirical segment risk $\bar{R}^\ell_t$ as defined in Eq. (14), and let $\mathcal{F}_\tau$ denote the filtration capturing all user histories, model predictions, and losses observed up to time $\tau$, then for any $\epsilon > 0$, we have:*

$$\Pr\left(\bar{R}^\ell_t - \mathbb{E}\left[\bar{R}^\ell_t \mid \mathcal{F}_{c^\ell_t - 1}\right] \geq \epsilon\right) \leq \exp\left(-2\epsilon^2 |\mathcal{W}^\ell_t|\right).$$

*Proof.* Let $X_\tau$ define a random variable that captures the surprise at time $\tau \in \mathcal{W}^\ell_t$, i.e.,

$$X_\tau := \mathcal{L}^{(\ell)}_\tau - \mathbb{E}\left[\mathcal{L}^{(\ell)}_\tau \mid \mathcal{F}_{\tau-1}\right], \tag{i}$$

where $\mathcal{L}^{(\ell)}_\tau(\mathcal{C}^\ell_{\lambda^\ell_\tau})$ is the observed loss, and the expectation is our best guess before time $\tau$.

We now define the cumulative sum over $X_\tau$ as:

$$S_k := \sum_{\tau=c^\ell_t}^{k} X_\tau, \quad \text{for } k \in [c^\ell_t, t]. \tag{ii}$$

Now, the sequence $\{S_k\}$ is a martingale with respect to the filtration $\mathcal{F}_k$. Specifically:

$$\mathbb{E}[S_k \mid \mathcal{F}_{k-1}] = S_{k-1}. \tag{iii}$$

This relation holds because:

$$S_k = S_{k-1} + X_k \quad \Rightarrow \quad \mathbb{E}[S_k \mid \mathcal{F}_{k-1}] = S_{k-1} + \mathbb{E}[X_k \mid \mathcal{F}_{k-1}].$$

Now,

$$\mathbb{E}[X_k \mid \mathcal{F}_{k-1}] = \mathbb{E}\left[\mathcal{L}^\ell_k - \mathbb{E}\left[\mathcal{L}^{(\ell)}_k \mid \mathcal{F}_{k-1}\right] \mid \mathcal{F}_{k-1}\right] \tag{iv}$$

By linearity and the idempotence of conditional expectation, we directly get:

$$\mathbb{E}[L^\ell_k \mid \mathcal{F}_{k-1}] - \mathbb{E}[L^\ell_k \mid \mathcal{F}_{k-1}] = 0. \tag{v}$$

Hence $X_k$ is a martingale difference, and $\{S_k\}$ is a martingale.

Also, since $\mathcal{L}^\ell_\tau(\mathcal{C}^\ell_{\lambda^\ell_\tau}) \in [0, 1]$, its conditional expectation also lies in $[0, 1]$, and therefore:

$$|X_k| \leq 1 \quad \text{i.e., the increments are bounded.}$$

Now, by Azuma–Hoeffding's inequality, for any martingale with bounded increments $|X_k| \leq 1$, the following holds: From Azuma–Hoeffding's inequality, we now have:

$$\Pr(S_t \geq \epsilon |\mathcal{W}^\ell_t|) \leq \exp\left(-2\epsilon^2 |\mathcal{W}^\ell_t|\right), \tag{vi}$$

where $\epsilon > 0$, and $|\mathcal{W}^\ell_t| = t - c^\ell_t + 1$.

Now we relate $S_t$ to the definition of empirical risk. Given the definition of average risk over a window, we have:

$$\mathbb{E}\left[\bar{R}^\ell_t \mid \mathcal{F}_{c^\ell_t - 1}\right] = \mathbb{E}\left[\frac{1}{w} \sum_{\tau=c^\ell_t}^{t} \mathcal{L}^\ell_\tau(\mathcal{C}^\ell_{\lambda^\ell_\tau}) \ \middle|\ \mathcal{F}_{c^\ell_t - 1}\right]$$

$$= \frac{1}{w} \sum_{\tau=c^\ell_t}^{t} \mathbb{E}\left[\mathcal{L}^\ell_\tau(\mathcal{C}^\ell_{\lambda^\ell_\tau}) \ \middle|\ \mathcal{F}_{c^\ell_t - 1}\right], \tag{vii}$$

where $w = |t - c_t^\ell + 1| := |\mathcal{W}_t^\ell|$.

Using the tower property of conditional expectation, for any $\tau \geq c_t^\ell$, we have:

$$\mathbb{E}\left[\mathcal{L}_\tau^\ell \mid \mathcal{F}_{c_t^\ell - 1}\right] = \mathbb{E}\left[\mathbb{E}\left[\mathcal{L}_\tau^\ell \mid \mathcal{F}_{\tau-1}\right] \mid \mathcal{F}_{c_t^\ell - 1}\right]. \tag{viii}$$

Now, given the expression for deviation from expected risk:

$$\bar{R}_t^\ell - \mathbb{E}[\bar{R}_t^\ell \mid \mathcal{F}_{c_t^\ell - 1}],$$

expanding this gives:

$$\frac{1}{w} \sum_{\tau=c_t^\ell}^t \mathcal{L}_\tau^\ell - \frac{1}{w} \sum_{\tau=c_t^\ell}^t \mathbb{E}\left[\mathcal{L}_\tau^\ell \mid \mathcal{F}_{\tau-1}\right].$$

Continuing from the previous expression, we now write:

$$\bar{R}_t^\ell - \mathbb{E}\left[\bar{R}_t^\ell \mid \mathcal{F}_{c_t^\ell - 1}\right] = \frac{1}{w} \sum_{\tau=c_t^\ell}^t \left(\mathcal{L}_\tau^\ell - \mathbb{E}[\mathcal{L}_\tau^\ell \mid \mathcal{F}_{\tau-1}]\right). \tag{ix}$$

Now applying the tower property again, and using the result from Eq. (iv), we observe:

$$\mathcal{L}_\tau^\ell - \mathbb{E}\left[\mathbb{E}[\mathcal{L}_\tau^\ell \mid \mathcal{F}_{\tau-1}] \mid \mathcal{F}_{c_t^\ell - 1}\right] = \mathbb{E}\left[X_\tau \mid \mathcal{F}_{c_t^\ell - 1}\right]. \tag{x}$$

Putting Eq. (x) into Eq. (ix), we obtain:

$$\mathbb{E}\left[\bar{R}_t^\ell - \mathbb{E}\left[\bar{R}_t^\ell \mid \mathcal{F}_{c_t^\ell - 1}\right]\right] = \frac{1}{w} \sum_{\tau=c_t^\ell}^t \mathbb{E}\left[X_\tau \mid \mathcal{F}_{c_t^\ell - 1}\right]. \tag{xi}$$

Since we are bounding this deviation in probability, we retain the raw form:

$$\bar{R}_t^\ell - \mathbb{E}\left[\bar{R}_t^\ell \mid \mathcal{F}_{c_t^\ell - 1}\right] = \frac{1}{w} \sum_{\tau=c_t^\ell}^t X_\tau = \frac{S_t}{w}. \tag{xii}$$

Now we finally substitute the result from Eq. (xii) into the Azuma–Hoeffding inequality Eq. (vi):

$$\Pr\left(\bar{R}_t^\ell - \mathbb{E}\left[\bar{R}_t^\ell \mid \mathcal{F}_{c_t^\ell - 1}\right] \geq \epsilon\right) = \Pr\left(\frac{S_t}{w} \geq \epsilon\right) \tag{xiii}$$

$$= \Pr\left(S_t \geq \epsilon w\right) \leq \exp\left(-2\epsilon^2 w\right). \tag{xiv}$$

Hence, we finally obtain the main result:

$$\boxed{\Pr\left(\bar{R}_t^\ell - \mathbb{E}[\bar{R}_t^\ell \mid \mathcal{F}_{c_t^\ell - 1}] \geq \epsilon\right) \leq \exp\left(-2\epsilon^2 \cdot |\mathcal{W}_t^\ell|\right)}$$

Hence Proved. $\qquad\qquad\qquad\qquad\qquad\qquad\qquad\qquad\qquad\qquad\qquad\qquad\qquad\qquad\qquad\qquad\qquad\qquad\square$

**Remark** Lemma 3 justifies using the empirical average risk $\bar{R}_t^\ell$ as a reliable proxy for the true conditional expectation and supports the adaptive threshold update rule in Eq. (15) of the framework.

**Corollary 1.** *Given the threshold update rule from Eq. (15) of the framework:* $\lambda_\ell^{t+1} = \lambda_\ell^t - \rho \left( \bar{R}_t^{(\ell)} - \alpha \right)$, *then for any* $\delta \in (0, 1)$*, with probability at least* $1 - \delta$*, the deviation of the update from the ideal update satisfies:*

$$\left| \lambda_\ell^{t+1*} - \lambda_\ell^{t+1} \right| := \rho \left| \mathbb{E}[\bar{R}_t^\ell \mid \mathcal{F}_{c_t^\ell - 1}] - \bar{R}_t^\ell \right| \leq \rho \cdot \sqrt{\frac{\log(1/\delta)}{2|\mathcal{W}_t^\ell|}}.$$

*Proof.* From Lemma 2, with probability at least $1 - \delta$, we have:

$$\bar{R}_t^\ell - \mathbb{E}\left[ \bar{R}_t^\ell \mid \mathcal{F}_{c_t^\ell - 1} \right] = \frac{S_t}{w} \quad \Rightarrow \quad \Pr\left( \bar{R}_t^\ell - \mathbb{E}[\bar{R}_t^\ell] \geq \epsilon \right) \leq \exp\left( -2\epsilon^2 w \right).$$

We now want to choose $\epsilon$ such that:

$$\exp\left( -2\epsilon^2 w \right) = \delta \quad \Rightarrow \quad \epsilon^2 = \frac{\log(1/\delta)}{2w} \quad \Rightarrow \quad \epsilon = \sqrt{\frac{\log(1/\delta)}{2w}} \tag{i}$$

Using Eq. (i), we can conclude that with probability at least $1 - \delta$:

$$\left| \bar{R}_t^\ell - \mathbb{E}[\bar{R}_t^\ell \mid \mathcal{F}_{c_t^\ell - 1}] \right| \leq \sqrt{\frac{\log(1/\delta)}{2w}}. \tag{ii}$$

Now substituting Eq. (ii) into the threshold update in framework's Eq. (15), and comparing with the ideal update:

$$\lambda_\ell^{t+1*} := \lambda_\ell^t - \rho \left( \mathbb{E}\left[ \bar{R}_t^\ell \mid \mathcal{F}_{c_t^\ell - 1} \right] - \alpha \right),$$

we conclude that:

$$\boxed{\left| \lambda_\ell^{t+1*} - \lambda_\ell^{t+1} \right| := \rho \left| \mathbb{E}[\bar{R}_t^\ell \mid \mathcal{F}_{c_t^\ell - 1}] - \bar{R}_t^\ell \right| \leq \rho \cdot \sqrt{\frac{\log(1/\delta)}{2|\mathcal{W}_t^\ell|}}.}$$

Hence Proved. □

**Remark** From Corollary 1 we observe that the adaptive threshold update remains close to its ideal value, even when using empirical segment risk. As the stable window length $|\mathcal{W}_t^\ell|$ increases, the deviation vanishes at a $O(1/\sqrt{|\mathcal{W}_t^\ell|})$ rate. This ensures the DAUO algorithm adapts reliably to user preferences over time, with provable statistical stability.

**Lemma 4.** *Let* $\mathcal{M}^1, \ldots, \mathcal{M}^L$ *be* $L$ *base models. Assume that for each model* $\mathcal{M}^\ell$*, the calibrated prediction set* $\mathcal{C}_{\lambda_\ell^t}^\ell$ *satisfies the per-model miss probability bound:* $\Pr\left( i_{\text{rel}}^{t+1}(u) \notin \mathcal{C}_{\lambda_\ell^t}^\ell(S_u^t) \mid \mathcal{F}_{t-1} \right) \leq \beta$ *for all* $\ell = 1, \ldots, L$*, where* $\beta := \frac{\alpha}{2} + \varepsilon$*, and* $\varepsilon := \sqrt{\frac{\log(4|\mathcal{U}|)}{2|\mathcal{U}|}}$*. Let* $\mathcal{C}_{\boldsymbol{\lambda}^t}^{\text{agg}}$ *denote the aggregated prediction set formed by randomized weighted majority voting, using aggregation weights* $\mathbf{w}^t \in \Delta^L$*, the probability simplex.*

*Then the miss probability of the ensemble satisfies:*

$$\Pr\left( i_{\text{rel}}^{t+1}(u) \notin \mathcal{C}_{\boldsymbol{\lambda}^t}^{\text{agg}} \mid \mathcal{F}_{t-1} \right) \leq \alpha + 2\varepsilon.$$

*Proof.* For any user $u$, we define the miss indicator for model $\mathcal{M}^\ell$ as:

$$M_\ell := \mathbf{1}\left\{ i_{\text{rel}}^{t+1}(u) \notin \mathcal{C}_{\lambda_\ell^t}^{(\ell)}(S_u^{(t)}) \right\}. \tag{i}$$

The ensemble predictor will fail if the true item receives insufficient support, i.e, the total weight of models that include the item is less than $\frac{1}{2}$. Equivalently, the total weight of models that miss the item exceeds $\frac{1}{2}$.

We formally define the total miss weight:

$$\sum_{\ell=1}^{L} w_\ell^t \cdot M_\ell. \tag{ii}$$

Then the ensemble misses if the above is $\geq \frac{1}{2}$. We wish to bound the probability of ensemble failure:

$$\Pr\left(\sum_{\ell=1}^{L} w_\ell^t \cdot M_\ell \geq \tfrac{1}{2} \,\middle|\, \mathcal{F}_{t-1}\right).$$

Applying Markov's inequality:

$$\Pr(X \geq a) \leq \frac{\mathbb{E}[X]}{a},$$

we obtain:

$$\Pr\left(\sum_{\ell=1}^{L} w_\ell^t \cdot M_\ell \geq \tfrac{1}{2} \,\middle|\, \mathcal{F}_{t-1}\right) \leq 2 \cdot \mathbb{E}\left[\sum_{\ell=1}^{L} w_\ell^t \cdot M_\ell \,\middle|\, \mathcal{F}_{t-1}\right]. \tag{iii}$$

Now, by linearity of expectation, we have:

$$\mathbb{E}\left[\sum_{\ell=1}^{L} w_\ell^t M_\ell \,\middle|\, \mathcal{F}_{t-1}\right] = \sum_{\ell=1}^{L} w_\ell^t \cdot \mathbb{E}\left[M_\ell \mid \mathcal{F}_{t-1}\right] = \sum_{\ell=1}^{L} w_\ell^t \cdot \Pr(M_\ell = 1 \mid \mathcal{F}_{t-1}). \tag{iv}$$

By Lemma 2, each model satisfies:

$$\Pr(M_\ell = 1 \mid \mathcal{F}_{t-1}) \leq \beta. \tag{v}$$

Therefore,

$$\sum_{\ell=1}^{L} w_\ell^t \cdot \Pr(M_\ell = 1 \mid \mathcal{F}_{t-1}) \leq \beta \cdot \sum_{\ell=1}^{L} w_\ell^t = \beta. \tag{vi}$$

Substituting result from Eq. (vi) to Eq. (iii) back, we get the final ensemble miss bound:

$$\boxed{\Pr\left(i_{\text{rel}}^{t+1}(u) \notin \mathcal{C}_{\boldsymbol{\lambda}^t}^{\text{agg}} \,\middle|\, \mathcal{F}_{t-1}\right) \leq 2\beta = \alpha + 2\varepsilon.} \tag{vii}$$

Hence Proved. □

***Remark*** Lemma 4 shows that the ensemble miss probability remains bounded by $\alpha + 2\varepsilon$ and preserves statistical validity despite possible correlation among predictors. As the calibration batch size $|\mathcal{U}| \to \infty$, the deviation $\varepsilon \to 0$, and the ensemble risk converges to $\alpha$.

### Proof of Theorem 2

*Proof.* Let $m := |\mathcal{U}|$ and $\varepsilon := \sqrt{\frac{\log(4m)}{2m}}$. Let $\mathcal{S} \subseteq \{1, \ldots, T\}$ denote the stable timestamps, where no preference change is detected, and let $\mathcal{D} := \{1, \ldots, T\} \setminus \mathcal{S}$ denote the detection delay rounds. Then, we can say:

$$|\mathcal{S}| = T - D_T, \quad |\mathcal{D}| = D_T.$$

From Lemmas 2 and 4 , the expected loss satisfies:

$$\mathbb{E}\left[\mathcal{L}_u\left(\mathcal{C}_{\boldsymbol{\lambda}^t}^{\text{agg}}\right) \,\middle|\, \mathcal{F}_{t-1}\right] \leq \alpha + 2\varepsilon. \tag{i}$$

For $t \in \mathcal{D}$, the DAUO algorithm may be out-of-calibration. We conservatively assume the worst-case loss of 1 at each such round. There are $D_T$ such rounds yielding:

$$\sum_{t \in \mathcal{D}} \mathbb{E}\left[\mathcal{L}_u^{(t)}\right] \leq D_T. \tag{ii}$$

Now we handle the additional slack from DKW failures. At each round $t \in [T]$ and for each model $\ell \in [L]$, we calibrate the threshold using DKW. So there are $T \times L$ calibration events.

Let $Z_{t,\ell} \in \{0, 1\}$ be the indicator that DKW calibration fails at round $t$ for model $\ell$.

Then the total number of failures is:

$$K := \sum_{t=1}^{T} \sum_{\ell=1}^{L} Z_{t,\ell}. \tag{iii}$$

By [Lemma 2](), each calibration failure has probability at most: $p := \frac{1}{2m}$. From Lemma 1, each DKW calibration failure has probability at most $p = \frac{1}{2m}$, and there are $T \times L$ such events. Thus, the expected number of failures is:

$$\mu := \mathbb{E}[K] = \frac{TL}{2m}.$$

We want to control the tail deviation:

$$\Pr(K \geq \mu + y) \leq \delta.$$

Using the Bernstein bound, we have:

$$\Pr(K \geq \mu + y) \leq \exp\left(\frac{-y^2}{2(\mu + y/3)}\right). \tag{iv}$$

To satisfy this inequality with probability $\geq 1 - \delta$, we choose $y$ to dominate both the average and tail slack. Following standard practice, we set:

$$y := \max\left\{\mu, \, 2\log\left(\frac{1}{\delta}\right)\right\}.$$

This guarantees:

$$\frac{y^2}{2(\mu + y/3)} \geq \log\left(\frac{1}{\delta}\right).$$

In realistic recommender settings, $m \gg L$, therefore:

$$\mu = \frac{TL}{2m} \leq 2\log\left(\frac{1}{\delta}\right).$$

Thus we may safely choose:

$$y = 2\log\left(\frac{1}{\delta}\right).$$

With this value, we get the high-probability bound:

$$K \leq \mu + y \leq \frac{TL}{2m} + 2\log\left(\frac{1}{\delta}\right). \tag{v}$$

Divide inequality (v) by $T$, we obtain:

$$\frac{K}{T} \leq \frac{TL}{2mT} + \frac{2\log(1/\delta)}{T}.$$

Since $\frac{TL}{2m} \leq 2\log(1/\delta)$ (by assumption), we get:

$$\frac{K}{T} \leq \frac{2\log(1/\delta)}{T}. \tag{vi}$$

Now combine the bounds from (i), (ii), and (vi):

$$\frac{1}{T}\sum_{t=1}^{T} \mathbb{E}\left[\mathcal{L}_u\left(\mathcal{C}_{\boldsymbol{\lambda}^t}^{\text{agg}}\right)\right] \leq \frac{T - D_T}{T}(\alpha + 2\varepsilon) + \frac{D_T}{T} \cdot 1 + \frac{K}{T}.$$

Substitute $\frac{K}{T} \leq \frac{2\log(1/\delta)}{T}$ and simplifying we get:

$$\frac{1}{T}\sum_{t=1}^{T}\mathbb{E}\left[\mathcal{L}_u\left(\mathcal{C}_{\boldsymbol{\lambda}^t}^{\mathrm{agg}}\right)\right] \leq \alpha + 2\varepsilon + \frac{D_T + 2\log(1/\delta)}{T}. \tag{vii}$$

At round $T+1$, the aggregated prediction set $\mathcal{C}_{\boldsymbol{\lambda}^T}^{\mathrm{agg}}$ is formed using the thresholds $\boldsymbol{\lambda}^T$ trained across rounds 1 to $T$.

Assuming no additional change-point occurs at round $T+1$, a standard assumption in horizon-end guarantees, the loss distribution is equivalent to a stable round. Thus, the same bound applies, yielding:

$$\boxed{\mathbb{E}_{u\sim\mathcal{U}}\left[\mathcal{L}_u(\mathcal{C}_{\boldsymbol{\lambda}^T}^{\mathrm{agg}})\right] \leq \alpha + 2\sqrt{\frac{\log(4|\mathcal{U}|)}{2|\mathcal{U}|}} + \frac{D_T + 2\log(1/\delta)}{T}.}$$

Hence Proved. □

# D. Implementation Details

In this section, we elaborate on the implementation details of the experiments conducted. The experiments were conducted on NVIDIA A40 GPU. Firstly, all base recommender models, NCF[19], CASER[39], SASRec[25], and FMLP-Rec[47] are trained for 100 epochs with a batch size of 256, a learning rate of 0.001, the Adam optimizer, and Binary Cross Entropy Loss (BCELoss). These models are implemented following their respective public repositories. User preference-aware baselines include TiSASRec[27], CDR[41], and Oracle4Rec[42]. TiSASRec extends SASRec with time-aware attention and relation-based temporal encoding, trained for 200 epochs with a batch size of 128. CDR employs a variational framework with domain-level disentanglement, trained for 200 epochs with a batch size of 512 and a learning rate of 0.0001. Oracle4Rec trains for 100 epochs with a batch size of 256 using a Transformer-style architecture with GELU activations and dropout regularization. These models retain their original optimization logic and regularization strategies. We furthermore implement three conformal prediction baselines: Split Conformal[40], EnbPI[43], and Online Conformal Prediction[1]. All conformal variants reuse the predicted score files from the base models and calculate expected loss based on ranking-based loss functions (e.g., MRR, NDCG, Recall). For Split Conformal, we determine the fixed prediction threshold via the $(1-\alpha)$-quantile of the first calibration timestamp, with $\alpha = 0.1$. For EnbPI, we use an ensemble of 10 bootstrapped recommendation models, with predictions aggregated using the sample mean. Prediction set widths were updated after each instance using a sliding window of the most recent $T = 5$ residuals. The miscoverage level was set to $\alpha = 0.1$, and expected loss was computed based on the same utility metrics. For Online Conformal Prediction, we use a decaying step size update rule, with the threshold updated after each instance. We set $\alpha = 0.1$ and used the same loss definitions as in other conformal methods explained above. The initial threshold $\lambda^0$ was shared across all conformal variants and our framework to ensure consistent initialization. Our proposed framework is implemented on top of the base recommendation model outputs. We conduct a manual search over the contrasting hyperparameters in our Bayesian change-point module: the shift sensitivity $\beta \in \{0.5, 0.7, 0.9, 1.1\}$ and the segment-length bias $\gamma \in \{0, 0.3, 0.5, 0.7, 1, 1.3, 1.5, 1.75, 2\}$. Based on manual validation of segment stability and calibration smoothness across datasets, we fixed $\beta = 0.7$ and $\gamma = 1.1$. The error tolerance value $\epsilon$ is chosen based on the dataset size and the confidence value $\delta$. The threshold update step size $\eta$ in Eq. (15) was set to 0.05 throughout. To ensure consistency and reproducibility, we reused the predicted score files generated by the trained base models for all conformal baselines and our framework.

### D.0.1. UTILITY FUNCTION DEFINITIONS

The user utility function $U_{metric}(i_{rel}^{t+1}, \mathcal{C}_{\lambda^t})$, used in the loss formulation in Eq. (5) in main paper quantifies how well the prediction set $\mathcal{C}_{\lambda^t} \subseteq \mathcal{I}$ captures the relevant item $i_{rel}^{t+1}$ under different evaluation metrics. We define the following instantiations of $U_{metric}$ based on standard recommendation metrics:

**Recall-based utility:**

$$U_{\mathrm{recall}}(i_{rel}^{t+1}, \mathcal{C}_{\lambda^t}) = \mathbb{I}[i_{rel}^{t+1} \in \mathcal{C}_{\lambda^t}]. \tag{viii}$$

This utility equals 1 if the relevant item is present in the prediction set and 0 otherwise.

**MRR-based utility:**

$$U_{\mathrm{mrr}}(i_{rel}^{t+1}, \mathcal{C}_{\lambda^t}) = \begin{cases} \frac{1}{r(i_{rel}^{t+1})}, & \text{if } i_{rel}^{t+1} \in \mathcal{C}_{\lambda^t}, \\ 0, & \text{otherwise}, \end{cases} \tag{ix}$$

where $r(i_{rel}^{t+1})$ denotes the rank position of the relevant item within $\mathcal{C}_{\lambda^t}$, assuming items are ordered by decreasing model score.

**NDCG-based utility:**

$$U_{\mathrm{ndcg}}(i_{rel}^{t+1}, \mathcal{C}_{\lambda^t}) = \frac{1}{\log_2(r(i_{rel}^{t+1}) + 1)} \cdot \mathbb{I}[i_{rel}^{t+1} \in \mathcal{C}_{\lambda^t}], \tag{x}$$

which discounts the gain based on the rank of the relevant item in the prediction set.

These definitions are used across all calibration and evaluation steps to compute utility-based loss values and coverage metrics.

# E. Detailed Experimentation Details

In the main paper, we introduced five different datasets to evaluate the effectiveness of our framework. Below, we provide further details on the datasets, data-preprocessing, the base models, the user-preference aware baselines, and the conformal baselines used for comparison.

## E.1. Datasets

- **Book-Crossing**[48]: a book-review dataset with explicit ratings and browsing logs.

- **Last.fm**[6]: music-streaming listening histories dataset providing implicit feedback.

- **Taobao**[22]: a large-scale e-commerce dataset with clicks, carts, and purchases attributes.

- **MovieLens**[18]: an explicit and implicit feedback dataset in the movie-rating domain.

- **Gowalla**[10]: a location-based social-network checkins dataset for point-of-interest recommendation.

All datasets are time-ordered, filtered using a 50-core strategy, and processed according to the data preprocessing and splitting procedure described below.

## E.2. Sampling and Data Splitting

- **Negative sampling.** Following the common experimentation strategy in recommendation frameworks, we select 50 non-interacted items per user at every time-stamp through negative sampling for training, validation, and testing.

- **Data Splitting.** Inspired by the sliding-window evaluation, we partition each dataset into five contiguous time-ordered batches $B_1, \ldots, B_5$ to capture potential shifts in user preferences over time. Within a batch, the first $80\%$ of interactions are used to train the model. The next $20\%$ are used to calibrate the conformal threshold $\lambda_\ell^t$ and weight parameters $\mathbf{w}^t$, while for the final interaction, the previously learned threshold and weight parameters are frozen and the framework is evaluated. The final results presented represent the average over all batches.

- **Multiple trials:** To account for variability in sampling, we repeat the experiments over 20 independent trials. For each trial, random negative samples were drawn for training, validation, and testing. The results were averaged across all the trials.

## E.3. Base Recommendation Models

We build our framework on top of four representative recommendation backbones, each capturing different modeling paradigms:

- **Neural Collaborative Filtering (NCF)**[19]: Involves combination of GMF (Generalized Matrix Factorization) with 8-dimensional embeddings and MLP using layers $[64, 32, 16]$ with ReLU and dropout; combined with a prediction layer over concatenated representations.

- **Caser**[39]: A convolutional sequence model using vertical and horizontal filters with varying receptive fields over a fixed-length user interaction sequence. Configured with embedding dimension $d = 50$, sequence length $L = 5$, number of horizontal and vertical filters $n_h = 16$, $n_v = 4$, followed by a fully connected layer and dropout ($p = 0.5$).

- **SASRec**[25]: A Transformer-style sequential recommender with 2 self-attention blocks, 1 attention head, hidden size of 50, max sequence length of 50, and dropout rate of 0.5. Layer normalization, residual connections, and position encoding are used to model sequential dependencies.

- **FMLP-Rec**[47]: A Filter-Enhanced MLP model replacing attention heads with learned convolutional filters. Configured with hidden size of 64, 2 filter-enhanced encoder layers, 2 attention heads, dropout $= 0.5$, and GELU activation. Position embeddings and layer normalization are applied on top of the input sequence.

## E.4. Preference-Aware Recommendation Models

To capture evolving user preferences and temporal context, we additionally incorporate three specialized preference-aware baselines:

- **TiSASRec:**[27] A time-aware sequential recommender model that extends SASRec by incorporating absolute and relative time information into the attention mechanism. We use 2 attention blocks, 1 attention head, and a hidden dimension of 50, along with a time matrix span of 256 and dropout rate of 0.2.

- **CDR (Causal Debiasing Recommendation):**[41] A user-centric causal recommendation model that disentangles user preferences across multiple training environments by learning group-invariant representations. We configure the MLP encoder as $[100, 20]$, preference encoder as $[100, 200]$, with latent variables all set to dimension 2. Dropout is set to 0.5 and batch norm is enabled.

- **Oracle4Rec:**[42] A a 5-layer Transformer-style encoder with hidden size 128, 2 attention heads, GELU activation, and dropout of 0.5. It learns forward-looking user preferences by leveraging future interactions as oracle guidance. It employs two parallel encoders with shared embeddings: a Past Information Encoder and a Future Information Encoder, each comprising a noise filtering module, a causal self-attention module, and an interaction prediction layer.

## E.5. Conformal Prediction Baselines

We implemented three conformal prediction baselines and adapted them for recommendation tasks using ranking-based losses based on recommendation metrics(Recall, MRR, and NDCG). For each method, we used calibrated scores and constructed dynamic prediction sets over time.

- **Split Conformal Prediction:** A simple offline baseline where a global threshold $\lambda$ is computed and fixed during calibration and inference. Prediction sets are constructed by thresholding sorted item scores per user. This method serves as a non-adaptive control with no online feedback or user preference modeling.

- **Ensemble Batch Prediction Interval (EnbPI):** A time series conformal approach adapted for sequential recommendation task, uses a chosen sliding window of size 5 and a shift size $s=1$ for full online behavior. An ensemble of 10 base models is used, and the prediction sets are constructed by aggregating top items across models using a mean-based ensemble score. The threshold $\lambda$ is updated after each interaction using decayed step size based on loss deviations.

- **Online Conformal:** A fully online adaptive approach that dynamically recalibrates the threshold $\lambda$ based on user-specific risk feedback. After each interaction, the conformal predictor computes the empirical loss based on the utility metric and updates $\lambda^t$ using a gradient-based rule with decay. Like EnbPI, prediction sets are constructed using sorted calibrated scores, but don't use model ensembling.

# F. Additional Experiments

## F.1. Results compared with base models and Preference-Aware baselines (Cont.)

We extend the analysis provided in the main paper, where we evaluate the CARE framework using four recommendation base models and against three user-preference-aware baselines in terms of recommendation metrics (i.e., MRR, Recall, NDCG). We present the results of the experimentations conducted on Taobao, MovieLens and Gowalla Datasets in Tables 5 and 6. These tables support the key findings: the CARE framework consistently controls risk within the predefined threshold $\alpha = 0.05$ with high confidence across all the base models, and as a result, it consistently outperforms all baselines on different performance metrics (MRR, Recall, NDCG) across datasets. This further validates the dataset-agnostic nature of our framework.

*Table 5.* Performance comparison of backbone recommenders (NeuMF, CASER, SASRec and FMLP-Rec) and preference-aware baselines (TiSASRec, CDR and Oracle4Rec) on **Taobao** and **MovieLens** using MRR, Recall@10, and NDCG@10. Bold indicates the best result, and underline indicates the second best.

| Method | Taobao | | | MovieLens | | |
|---|---|---|---|---|---|---|
| | MRR ↑ | Recall ↑ | NDCG ↑ | MRR ↑ | Recall ↑ | NDCG ↑ |
| NeuMF | 0.275 | 0.556 | 0.289 | 0.342 | 0.721 | 0.358 |
| NeuMF + CARE (Ours) | 0.292 | 0.587 | 0.298 | 0.356 | 0.739 | 0.368 |
| CASER | 0.320 | 0.589 | 0.338 | 0.381 | 0.775 | 0.389 |
| CASER + CARE (Ours) | 0.343 | 0.612 | 0.350 | 0.391 | 0.798 | 0.395 |
| SASRec | 0.337 | 0.605 | 0.338 | 0.395 | 0.795 | 0.405 |
| SASRec + CARE (Ours) | 0.353 | 0.625 | 0.359 | 0.413 | 0.807 | 0.423 |
| FMLP-Rec | 0.363 | 0.612 | 0.361 | 0.405 | 0.811 | 0.415 |
| FMLP-Rec + CARE (Ours) | **0.373** | **0.649** | **0.385** | **0.435** | **0.851** | **0.454** |
| **Preference-Aware Models** | | | | | | |
| TiSASRec | 0.348 | 0.610 | 0.353 | 0.402 | 0.802 | 0.412 |
| CDR | 0.339 | 0.609 | 0.351 | 0.399 | 0.795 | 0.405 |
| Oracle4Rec | 0.363 | 0.615 | 0.363 | 0.411 | 0.835 | 0.419 |

## F.2. Results compared to Conformal baselines (Cont.)

Next, we continue our analysis comparing our framework with different conformal baselines in terms of coverage and set size. We conduct the experiments on Last.fM (Table 7), Taobao (Table 8), MovieLens (Table 9) and Gowalla (Table 10) datasets respectively and compare the results on base recommender models. The results reaffirm the main paper observations that our framework can ensure the best coverage–efficiency trade-off on every base model across datasets, ensuring valid recommendation sets.

## F.3. Parameter Analysis

We analyze the influence of error rate $\alpha$, confidence parameter $\delta$, change-point detector parameters $(\beta, \gamma)$, and the number of experts $L$ on the recommendation sets generated by the CARE framework.

We first evaluate the impact of error rate $\alpha$, varying in $[0.05, 0.07, 0.10, 0.12, 0.15]$, on performance and the average prediction set sizes under fixed confidence thresholds $\delta = 0.05$ using the Book-Crossing dataset. As shown in Figure 3, as the error rate $\alpha$ increases, the performance across different metrics (MRR, Recall, NDCG) as well as the average set size across all models decreases. This decreasing trend demonstrates the framework's ability to generate valid prediction sets that adapt to the error rate $\alpha$.

We further evaluate the effect of varying confidence $\delta \in [0.05, 0.10, 0.15, 0.20, 0.25]$ on performance and average set sizes under fixed risk thresholds ($\alpha = 0.07$) using the Last.fm dataset in Figure 4. In general, all the models show a decreasing trend, validating the effectiveness of the framework. This is because relaxing confidence in risk constraints makes predictions less conservative, thereby reducing the number of items included in the recommendation set. Interestingly, performance and

*Table 6.* Performance comparison of backbone recommenders (NeuMF, CASER, SASRec and FMLP-Rec) and preference-aware baselines (TiSASRec, CDR and Oracle4Rec) on **Gowalla** using MRR, Recall@10, and NDCG@10. Bold indicates the best result, and underline indicates the second best.

| Method | MRR ↑ | Recall ↑ | NDCG ↑ |
|---|---|---|---|
| NeuMF | 0.286 | 0.565 | 0.289 |
| NeuMF + CARE (Ours) | 0.291 | 0.577 | 0.309 |
| CASER | 0.322 | 0.589 | 0.336 |
| CASER + CARE (Ours) | 0.334 | 0.602 | 0.343 |
| SASRec | 0.332 | 0.599 | 0.349 |
| SASRec + CARE (Ours) | 0.344 | 0.612 | 0.359 |
| FMLP-Rec | 0.342 | 0.605 | 0.355 |
| FMLP-Rec + CARE (Ours) | **0.359** | **0.632** | **0.364** |
| **Preference-Aware Models** | | | |
| TiSASRec | 0.339 | 0.601 | 0.350 |
| CDR | 0.333 | 0.595 | 0.349 |
| Oracle4Rec | 0.343 | 0.609 | 0.360 |

*Table 7.* Comparison in terms in terms of coverage and average prediction set size with conformal baselines (Split Conformal, EnbPI and Online Conformal) evaluated on four base recommenders (NeuMF, CASER, SASRec, and FMLP-Rec) using the **Last.fM** dataset. The error rate is set as $\alpha = 0.10$. Bold indicates the best result, underline indicates the second best.

| Base Model | Coverage ↑ | | | | Set Size ↓ | | | |
|---|---|---|---|---|---|---|---|---|
| | Split | EnbPI | Online | CARE (Ours) | Split | EnbPI | Online | CARE (Ours) |
| NeuMF | 0.833 | 0.858 | 0.881 | 0.901 | 41 | 42 | 42 | 43 |
| CASER | 0.835 | 0.868 | 0.884 | 0.903 | 40 | 42 | 43 | 41 |
| SASRec | 0.849 | 0.870 | 0.889 | 0.905 | 40 | 41 | 42 | 40 |
| FMLP-Rec | 0.855 | 0.873 | 0.899 | **0.907** | 40 | 40 | 40 | **39** |

*Table 8.* Comparison in terms in terms of coverage and average prediction set size with conformal baselines (Split Conformal, EnbPI and Online Conformal) evaluated on four base recommenders (NeuMF, CASER, SASRec, and FMLP-Rec) using the **Taobao** dataset. The error rate is set as $\alpha = 0.10$. Bold indicates the best result, underline indicates the second best.

| Base Model | Coverage ↑ | | | | Set Size ↓ | | | |
|---|---|---|---|---|---|---|---|---|
| | Split | EnbPI | Online | CARE (Ours) | Split | EnbPI | Online | CARE (Ours) |
| NeuMF | 0.828 | 0.859 | 0.880 | 0.901 | 42 | 43 | 44 | 44 |
| CASER | 0.835 | 0.862 | 0.881 | 0.903 | 42 | 43 | 42 | 42 |
| SASRec | 0.836 | 0.871 | 0.900 | 0.909 | 41 | 42 | 42 | 41 |
| FMLP-Rec | 0.838 | 0.879 | 0.901 | **0.911** | 41 | 41 | 41 | **40** |

set sizes show a smaller decline for $\delta$ compared to $\alpha$, since $\delta$ controls only the confidence with which the risk constraint must hold i.e., the probability mass in the extreme tail, whereas $\alpha$ sets the risk level itself.

We also perform a grid study of the change-point parameters $\beta$ (shift sensitivity) and $\gamma$ (segment-length prior) on Book-Crossing dataset while holding all other settings fixed. Table 11 reports average set size / coverage. We observe a consistent trade-off: larger $\beta$ or smaller $\gamma$ makes the detector more responsive, yielding slightly larger sets with improved coverage; the reverse favors tighter sets but risks transient under-coverage. In practice, we set $\beta$=0.7, $\gamma$=1.1 as a balanced choice across datasets. Finally, we vary the number of bootstrapped experts $L \in \{5, 10, 20\}$ and observe that CARE's set size and coverage are stable (Table 12). This empirical insensitivity is consistent with Theorem 1, which implies only a $\mathcal{O}(\sqrt{\ln L})$ growth term in the ensemble set size bound.

Overall, this parameter analysis guides real-world applications in balancing performance and recommendation set compact-

*Table 9.* Comparison in terms in terms of coverage and average prediction set size with conformal baselines (Split Conformal, EnbPI and Online Conformal) evaluated on four base recommenders (NeuMF, CASER, SASRec, and FMLP-Rec) using the **MovieLens** dataset. The error rate is set as $\alpha = 0.10$. Bold indicates the best result, underline indicates the second best.

| Base Model | Coverage ↑ | | | | Set Size ↓ | | | |
| --- | --- | --- | --- | --- | --- | --- | --- | --- |
| | Split | EnbPI | Online | CARE (Ours) | Split | EnbPI | Online | CARE (Ours) |
| NeuMF | 0.851 | 0.859 | 0.862 | 0.901 | 39 | 40 | 40 | 39 |
| CASER | 0.861 | 0.878 | 0.872 | 0.901 | 39 | 40 | 40 | 38 |
| SASRec | 0.867 | 0.881 | 0.891 | **0.902** | 38 | 38 | 39 | 36 |
| FMLP-Rec | 0.871 | 0.889 | 0.901 | 0.901 | 38 | 37 | 38 | **35** |

*Table 10.* Comparison in terms in terms of coverage and average prediction set size with conformal baselines (Split Conformal, EnbPI and Online Conformal) evaluated on four base recommenders (NeuMF, CASER, SASRec, and FMLP-Rec) using the **Gowalla** dataset. The error rate is set as $\alpha = 0.10$. Bold indicates the best result, underline indicates the second best.

| Base Model | Coverage ↑ | | | | Set Size ↓ | | | |
| --- | --- | --- | --- | --- | --- | --- | --- | --- |
| | Split | EnbPI | Online | CARE (Ours) | Split | EnbPI | Online | CARE (Ours) |
| NeuMF | 0.829 | 0.851 | 0.871 | 0.901 | 43 | 43 | 44 | 44 |
| CASER | 0.831 | 0.860 | 0.883 | 0.902 | 43 | 42 | 44 | 43 |
| SASRec | 0.837 | 0.870 | 0.895 | 0.901 | 43 | 42 | 43 | 42 |
| FMLP-Rec | 0.842 | 0.875 | 0.900 | **0.905** | 42 | 47 | 43 | **41** |

ness with confidence guarantees.

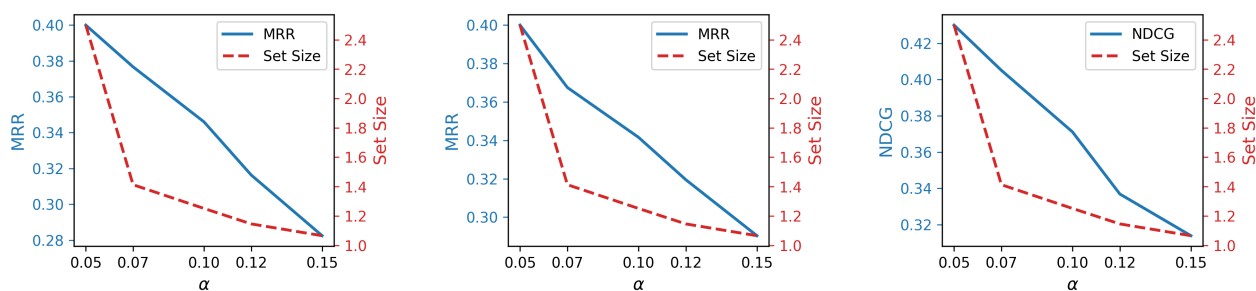

*Figure 3.* Performance analysis on the **Book-Crossing** dataset for varying $\alpha \in 0.05, 0.07, 0.10, 0.12, 0.15$ with fixed $\delta = 0.05$, shown in terms of recommendation metrics and prediction set size.

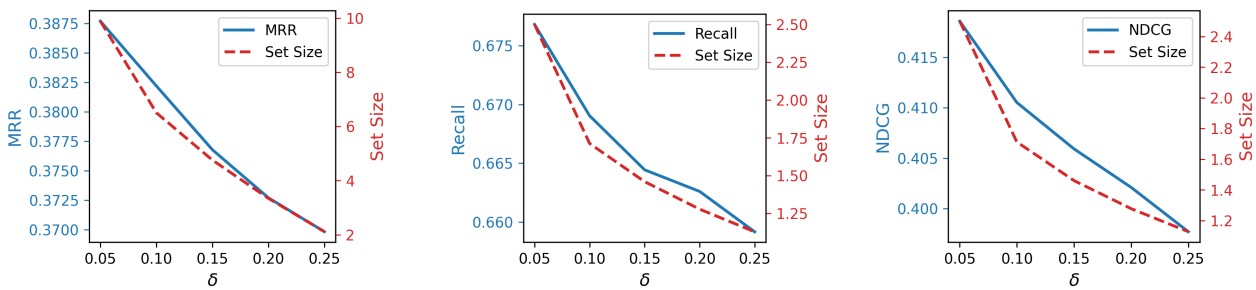

*Figure 4.* Performance analysis on the **Last.fm** dataset for varying $\delta \in 0.05, 0.1, 00.15, 0.20, 0.255$ with fixed $\alpha = 0.07$, shown in terms of recommendation metrics and prediction set size.

*Table 11.* Set size (left) and coverage (right) for different $\gamma$ and $\beta$ on **Book-Crossing** Dataset.

| $\gamma \downarrow$ / $\beta \rightarrow$ | 0.5 | 0.7 | 1.0 |
|---|---|---|---|
| 0.9 | 44.6 / 0.920 | 45.6 / 0.924 | 46.5 / 0.930 |
| 1.1 | 42.1 / 0.895 | 42.8 / 0.908 | 43.5 / 0.912 |
| 1.3 | 41.2 / 0.889 | 42.2 / 0.892 | 43.1 / 0.901 |

*Table 12.* Robustness to ensemble size $L$ on **Book-Crossing** Dataset (set size / coverage).

| $L$ | 5 | 10 | 20 |
|---|---|---|---|
| set size / coverage | 42.5 / 0.906 | 42.9 / 0.908 | 43.5 / 0.908 |

## F.4. Ablation Study

To evaluate the effect of the two detection components in CARE, we perform an ablation study by selectively removing each loss-based non-stationarity term. We follow the same experimental protocol as described in Section E.2, with the error rate fixed at $\alpha = 0.1$ and confidence level $\delta = 0.05$. We report results on the Book-Crossing dataset with the SASRec backbone, and analyze the performance in terms of a) *validity*: measured as realized coverage against the error rate, b) *compactness*: measured in terms of the average set size, and c) *robustness*: which is measured in terms of the recommendation set volatility across the time stamps. We define the robustness parameter $\chi$ as:

$$\chi \;=\; \frac{1}{T-1} \sum_{t=2}^{T} \frac{\left| \mathcal{C}_{\boldsymbol{\lambda}^t}^{\mathrm{agg}} \, \Delta \, \mathcal{C}_{\boldsymbol{\lambda}^{t-1}}^{\mathrm{agg}} \right|}{\left| \mathcal{C}_{\boldsymbol{\lambda}^{t-1}}^{\mathrm{agg}} \right|},$$

where $\Delta$ denotes the difference between consecutive aggregated prediction sets.

We consider he following cases: (1) **w/o** $d_\ell^{\mathrm{ldd}}$, where only the concept-sensitive divergence $d_\ell^{\mathrm{con}}$ is retained; and (2) **w/o** $d_{\mathrm{con}}$, where only the loss discrepency distance $d_{\mathrm{ldd}}$ is retained. The results are denoted in Table 13. The results lead to following key observations:

*Table 13.* Ablation of detection components on **Book-Crossing** Dataset

| Variant | Coverage ↑ | Avg set size ↓ | Volatility $\chi$ ↓ |
|---|---|---|---|
| CARE | 0.908 | 42.8 | 0.12 |
| w/o $d_\ell^{\mathrm{ldd}}$ | 0.872 | 44.7 | 0.10 |
| w/o $d_\ell^{\mathrm{con}}$ | 0.907 | 43.1 | 0.23 |

- Firstly, removing $d_\ell^{\mathrm{ldd}}$ substantially reduces validity. The coverage drops below the target $\alpha$. As a result, the framework tries to compensate by inflating the prediction sets. This is because, without the loss-discrepancy term, the detector becomes insensitive to uniform increases in difficulty across models. In such cases, shifts that affect all experts simultaneously go undetected, and calibration lags behind, leading to systematic under-coverage.

- Secondly, removing $d_\ell^{\mathrm{con}}$ primarily degrades robustness. Although coverage remains close to the target and the average set size looks competitive, the volatility $\chi$ nearly doubles. This indicates unstable calibration as the threshold $\lambda$ fluctuates sharply in response to transient expert disagreements, even when the underlying distribution is relatively stable. In practice, this results in inconsistent recommendation sets from one time step to the next, potentially harming user trust.

- Finally, the full CARE framework, by jointly utilizing both the loss-discrepancy and the concept-sensitive terms, balances the strengths of each detector. The loss-discrepancy term guards against systematic difficulty shifts, while the concept-sensitive term dampens volatility caused by transient expert fluctuations. Their combination ensures that coverage stays close to the nominal target (validity), prediction sets remain as small as possible without sacrificing risk guarantees (efficiency), and threshold updates evolve smoothly over time (robustness).

*Table 14.* Performance under varying drift intensities with abrupt shifts on Book-Crossing using the SASRec backbone. Target coverage is 0.90. Detection delay $D_T$ is measured in interaction steps.

| Drift Intensity | Split | | | EnbPI | | | Online Conformal | | | CARE (Ours) | | |
| --- | --- | --- | --- | --- | --- | --- | --- | --- | --- | --- | --- | --- |
| | Cov.↑ | Size↓ | $D_T \downarrow$ | Cov.↑ | Size↓ | $D_T \downarrow$ | Cov.↑ | Size↓ | $D_T \downarrow$ | Cov.↑ | Size↓ | $D_T \downarrow$ |
| Low (Cosine 0.8–0.9) | 0.881 | 44.7 | N/A | 0.889 | 44.5 | N/A | 0.898 | 44.1 | N/A | 0.905 | 43.5 | 8.2 |
| Medium (Cosine 0.4–0.6) | 0.842 | 43.2 | N/A | 0.867 | 47.8 | N/A | 0.881 | 47.2 | N/A | 0.903 | 44.1 | 3.8 |
| High (Cosine 0.0–0.2) | 0.765 | 42.9 | N/A | 0.821 | 52.3 | N/A | 0.865 | 54.8 | N/A | 0.908 | 45.6 | 1.5 |

*Table 15.* Performance under abrupt versus gradual shifts on Book-Crossing using the SASRec backbone. Medium drift intensity is used. The gradual shift occurs over a transition window of $W = 15$ steps. Target coverage is 0.90. Detection delay $D_T$ is measured in interaction steps.

| Shift Speed | Split | | | EnbPI | | | Online Conformal | | | CARE (Ours) | | |
| --- | --- | --- | --- | --- | --- | --- | --- | --- | --- | --- | --- | --- |
| | Cov.↑ | Size↓ | $D_T \downarrow$ | Cov.↑ | Size↓ | $D_T \downarrow$ | Cov.↑ | Size↓ | $D_T \downarrow$ | Cov.↑ | Size↓ | $D_T \downarrow$ |
| Abrupt | 0.842 | 43.2 | N/A | 0.867 | 47.8 | N/A | 0.881 | 47.2 | N/A | 0.903 | 44.1 | 3.8 |
| Gradual | 0.861 | 42.9 | N/A | 0.875 | 46.2 | N/A | 0.889 | 45.8 | N/A | 0.906 | 43.8 | 11.2 |

These results show that each component is complementary and addresses a distinct failure mode, and together they form a balanced and reliable detector of preference changes. Hence, both signals are indispensable for achieving stable uncertainty-aware recommendations under non-stationary user behavior.

### F.5. Semi-Synthetic Drift Test

To evaluate CARE under controlled non-stationarity, we conduct a semi-synthetic drift stress test on the Book-Crossing dataset using the SASRec backbone. Since in real-world recommendation datasets, we do not have ground-truth preference-shift timestamps, we inject preference shifts at known timestamps by constructing proxy users with controlled similarity to the original user profile. Drift intensity is controlled by cosine similarity between the original and shifted preference profile, where, low drift corresponds to cosine similarity in $[0.8, 0.9]$, medium drift to $[0.4, 0.6]$, and high drift to $[0.0, 0.2]$. We evaluate both abrupt shifts and gradual shifts; for gradual shifts, the transition occurs over a window of $W = 15$ interaction steps. The target coverage is fixed at 0.90, and the detection delay $D_T$ is measured in interaction steps.

Table 14 reports performance under abrupt shifts with varying drift intensity. CARE is the only method that consistently maintains coverage at or above the target level across all drift intensities, achieving coverage values of 0.905, 0.903, and 0.908 under low, medium, and high drift, respectively. The detection delay decreases from 8.2 to 3.8 and then to 1.5 steps as drift intensity increases, indicating that stronger shifts provide clearer evidence for the change-point detector and are localized more quickly. In contrast, the conformal baselines fall below the desired coverage level, especially under high drift, where their coverage drops substantially. Table 15 compares abrupt and gradual shifts at medium drift intensity. CARE maintains valid coverage in both cases, with coverage of 0.903 under abrupt shifts and 0.906 under gradual shifts, while keeping the average set size nearly unchanged. The detection delay is larger for gradual shifts, increasing from 3.8 to 11.2 steps, which is expected because gradual transitions produce weaker per-step evidence of distributional change. Nevertheless, CARE preserves the target coverage, showing that the proposed adaptive recalibration remains reliable even when preference shifts happen slowly.

## G. Intuition of Adaptive Dynamics in CARE

To provide an intuitive understanding of the CARE framework's adaptive capability, we visualize the internal dynamics of the DAUO algorithm during a user session based on interactions from the Taobao dataset, designed to illustrate a sequence of preference changes. Figure 5 describes how the three key variables evolve: the rolling risk, the calibration threshold ($\lambda$), and the prediction set size. Figure 5 illustrates the sequence of the adaptation loop. Initially, stable user behavior allows for a high threshold ($\lambda \approx 0.62$) and compact set size. A sudden preference changes degrades the ranking quality, causing a risk spike. The DAUO update rule (Eq. 15) counters this by lowering $\lambda$ (Middle Panel), which accordingly expands the prediction set (Bottom Panel) to restore coverage. Notably, the set size stabilizes at a higher level rather than returning to baseline because the underlying backbone model remains frozen. CARE correctly identifies that the frozen model is now less accurate for the new user preference and permanently maintains a larger safety margin to ensure continued risk control.

# H. Discussion

Our framework CARE reframes sequential recommendation as an uncertainty-aware prediction set problem that (1) hedges an ensemble of bootstrapped recommenders through Hedge weighting with adaptive conformal thresholds, (2) detects user-specific preference changes without any heuristically chosen window lengths utilizing a Bayesian changepoint detection model, and (3) provides sample guarantees that both the expected set size and the utility-based risk stay near-optimal under non-stationary preferences. Our claims are empirically supported as CARE consistently outperforms base recommender models and preference-aware recommender baselines on various recommendation metrics while maintaining tight and valid $(1 - \alpha)$ coverage across five public datasets. It does so without adding any significant training time, hence it can be expanded to recent popular generative models (Rajput et al., 2023; Zhai et al., 2024; Deng et al., 2025; Han et al., 2025). It is also robust in addressing broader concerns raised in the recommendations. Because thresholds and ensemble weights are updated externally with respect to a platform-defined utility function $U_{metric}$, the framework can incorporate fairness- or diversity-aware objectives directly. For example, $U_{metric}$ can be defined to penalize concentration or unsafe content, or combined with exposure caps and pre-filters; the coverage guarantees then hold with respect to this modified $U_{metric}$, requiring no change to the theory. This flexibility ensures resilience to issues such as filter bubbles or echo chambers. Different fairness definitions across user groups is also supported by the mechanism. Since thresholds and exponential-weights are updated externally, calibration can be performed separately for groups (e.g., by demographics, region, or activity level). Replacing $|\mathcal{U}|$ with $|\mathcal{U}_g|$ yields valid guarantees for each group independently, preserving equitable coverage across heterogeneous populations. Users in smaller or sparser cohorts may see slightly larger average set sizes due to finite-sample slack, but validity is preserved as shown in Theorem 1 and Theorem 2.

CARE does face the finite-sample effect. While the smaller calibration size continues ensuring the validity in a dynamic environment, it may lead to more conservative prediction sets as shown in our theoretical results. Also, as commonly seen in conformal strategies, CARE can only be as good as the confidence scores it calibrates. If a backbone recommender produces poorly ranked logits with poorly calibrated backbones (NeuMF), CARE's sets are ~15% larger than with stronger models (FMLP-Rec). We aim to address these challenges in future work. Overall, our work bridges the gap between sequential recommender systems' lack of reliability in adaptive environments with changing user preferences, which is a pragmatic step towards inspiring future research in trustworthy recommendation systems.

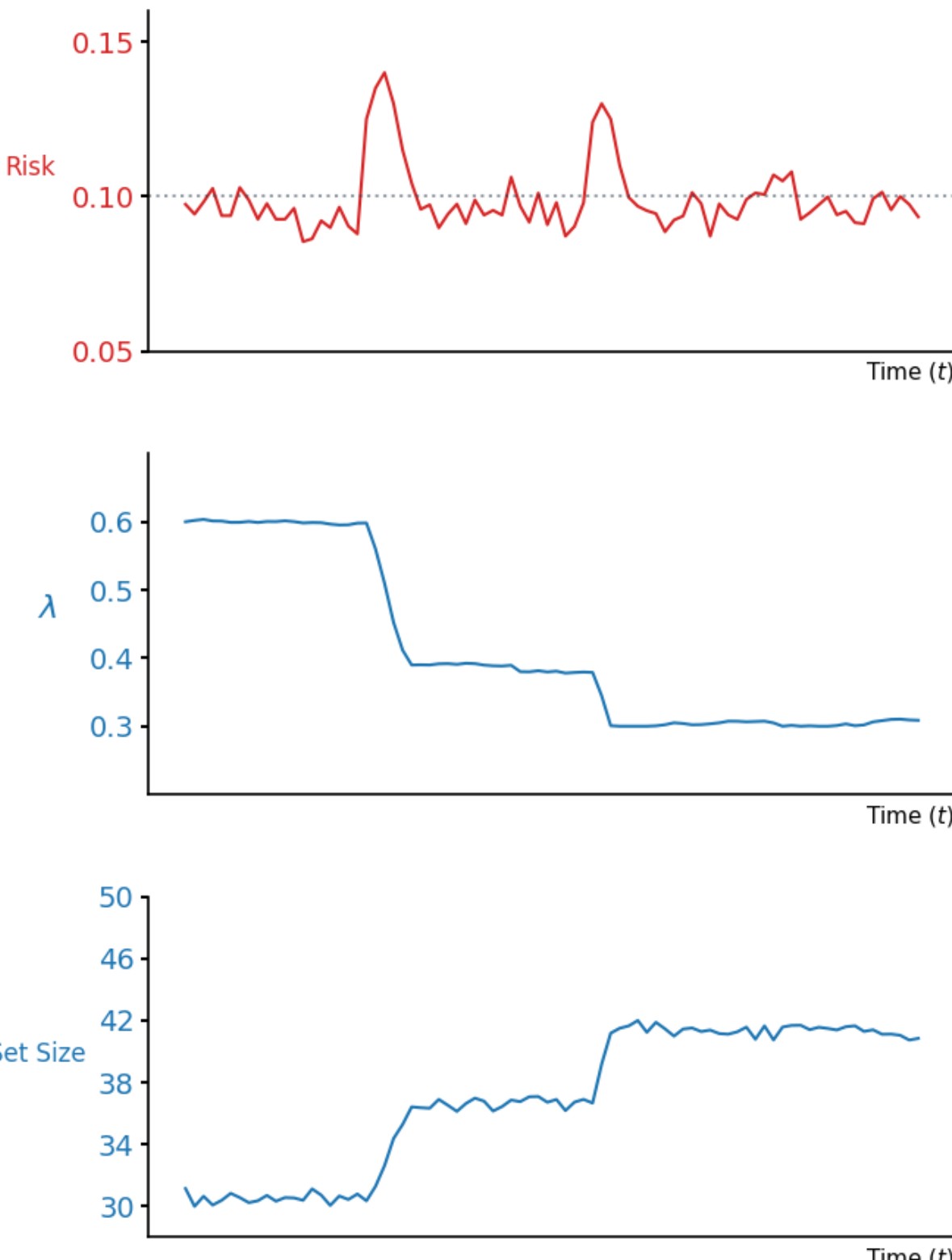

*Figure 5.* **Dynamic Adaptation of CARE under User Preference Changes.** (Top) The rolling risk spikes above the target $\alpha = 0.10$, indicating preference changes. (Middle) The calibration threshold $\lambda^t$ reacts immediately by lowering (Eq. 15) to loosen constraints. (Bottom) The prediction set size accordingly increases, confirming the framework's ability to actively detect and adapt to preference changes in real-time.

