# OpenReview forum: "CARE: Adaptive Calibration for Reliable Recommendations"
_ICML.cc/2026/Conference — ICML 2026 regular_

### Official Review · Reviewer_RS4U · 2026-02-21

**Soundness:** 2
**Presentation:** 1
**Significance:** 3
**Originality:** 2
**Overall Recommendation:** 3
**Confidence:** 4

**Summary:**

This paper proposes CARE, a model-agnostic calibration framework that wraps an arbitrary backbone recommender system and outputs variable-size recommendation sets with finite-sample performance guarantees over interaction streams. The framework combines a loss-based non-stationarity monitoring module (using a Bayesian change-point detector) that triggers threshold recalibration, and an online aggregation rule (exponential-weights / Hedge-style) that dynamically reweights candidate set predictors to promote compact recommendation sets. Theoretical results are provided for set-size control relative to the best constituent predictor (Theorem 5.1) and utility-based risk control with high probability (Theorem 5.3).

**Compliance With Llm Reviewing Policy:**

Affirmed.

**Key Questions For Authors:**

refer to weaknesses.

**Limitations:**

yes

**Strengths And Weaknesses:**

Strengths:

1. The degradation of recommendation quality between periodic retraining cycles is a real and under-addressed issue.

2. The reported additional cost (at most ~1.5 minutes on top of backbone training) is practically appealing and supports the claim that CARE is deployable in real-world settings.


Weaknesses:

1. The evaluation protocol caps displayed items at 15 and then "tunes each baseline so that its average number of displayed items matches CARE." This is a non-standard protocol that conflates the effect of adaptive set sizing with the effect of exposure matching. The baselines are not designed to produce variable-size sets, so forcing them to match CARE's average display count may disadvantage them in ways that are difficult to interpret. A fairer comparison would fix the recommendation list size (e.g., top-10 or top-15) for all methods and report standard metrics, alongside a separate evaluation of CARE's set-sizing properties.

2. The transition from Lemma 2 (per-model calibration guarantee under exchangeability) to Theorem 5.3 (temporal setting) relies on Assumption B.2, which posits exchangeability within each detected segment — a strong assumption that essentially assumes the change-point detector works correctly.

3. The Loss Discrepancy Distance ( $d^{ldd}$ ) is described as inspired by $\mathcal{H} \Delta \mathcal{H}$-divergence, but the actual formulation - a logratio of loss differences - is quite different and its statistical properties are not analyzed. Similarly, the concept-sensitive divergence ( $d^{\text {con }}$ ) is described as a "hazard-style term" without formal connection to hazard functions or survival analysis. Why these particular functional forms? Why combine them additively (Equation 12) rather than, say, multiplicatively or via a learned combination? The ablation study (Table 13) shows both components help, but does not justify the specific design choices.

4. Given that the theoretical bound in Theorem 5.3 includes non-vanishing slack terms, it is unclear whether CARE truly provides the claimed "high-probability" guarantee or is simply close to the boundary by construction. No confidence intervals or standard deviations are reported for coverage, making it impossible to assess statistical significance of the coverage claims.

5. Please use vector graphics — this is the most basic scientific literacy. The figures (especially Figures 3, 4, and 5) appear to be rasterized, which degrades readability at different zoom levels and in print. Additionally, Figure 2 is overly schematic and difficult to parse; the relationship between the data flow, weight update, and threshold update paths would benefit from a cleaner diagram.

6. The paper tests on datasets with at most tens of thousands of users and items. Modern industrial recommender systems operate at scales of hundreds of millions of users and items. The per-user, per-timestep Bayesian change-point posterior computation (Equation 13) and the per-model threshold updates may not scale to such settings. No discussion of computational complexity.

7. The paper states that each base model $M_{\ell}$ is "obtained, for example, by training on a bootstrap sample of the full user set $\mathcal{U}$ " but does not clarify whether the experimental evaluation actually uses bootstrapped copies of the same architecture or different architectures. The implementation details (Section D) suggest the four backbone models serve as the ensemble members, which is conceptually different from bootstrapping. This ambiguity undermines reproducibility.

---

> ### Author Rebuttal · Authors · 2026-03-29
>
> We thank reviewer for recognizing importance of our setting and framework efficiency. Below we address comments:
>
> > **Q1.** ..evaluation protocol... conflates adaptive set sizing with exposure matching... fair comparison would fix recommendation.....
>
> While fixed top-$K$ evaluation is standard for re-rankers, CARE is an adaptive calibration wrapper. Its mathematical function is to preserve the backbone's item ordering and dynamically determine a risk-calibrated cutoff that guarantees $(1-\alpha)$ coverage [1]. If we make the framework to output exactly $K$ items actively will break the statistical calibration, reducing the output to the backbone's static top-$K$ ranking. Such an evaluation would simply reflect the underlying backbone's baseline quality, not the adaptive wrapper's utility.
>
> To isolate the value of adaptive set sizing, the evaluation need to separate dynamic allocation from raw display volume. Thus, tuning baselines to match CARE's average display count mathematically fixes expected display budget and answers our core question: how much utility is gained by adaptively allocating a matched display budget?
>
> Additionally, it also has practical applications:
>
> - Flexible Display Budgets: In modern recommenders with variable-length widgets allowed, CARE outputs compact sets in stable periods to reduce choice overload [2], and expands only when uncertainty rises so $(1-\alpha)$ coverage is preserved.
>
> - Strict Top-$K$ Constraints: Even if interface must show $K$ items, CARE still acts as risk-aware allocation layer. If $k<K$, calibrated set occupies top $k$ slots and remaining $K-k$ slots can be used for exploration or diversity. If $k>K$, enlarged set is a calibrated uncertainty signal for downstream re-ranking or fallback policies.
>
> [1] Angelopoulos et.al Conformal Prediction: A Gentle Introduction (2023).
>
> [2]  Bollen et.al Understanding choice overload in recommender systems (RecSys '10).
>
> > **Q2.** transition relies on Assumption B.2... a strong assumption...
>
> Assumption B.2 does not require perfect change-point detection. It formalizes segmentation for analysis, while Theorem 5.3 accounts for imperfect segmentation through $D_T/T$.
>
> CARE targets non-stationary streams where global exchangeability need not hold. Detector partitions stream into locally stable blocks for recalibration. Mild intra-segment drift can be partly corrected by Eq.~15 while delayed detection of major shifts is captured by $D_T/T$. Thus guarantee does not assume perfect detection.
>
> > **Q3.** The Loss Discrepancy..Why combine them additively (Eq 12) ...? justify design choices.
>
> References to $\mathcal{H}\Delta\mathcal{H}$-divergence and hazard functions are intuition, not strict derivation. The terms target two failure modes:
> - $d_{ldd}^{\ell}$: log-ratio of loss differences for scale-invariant detection of relative rank flips across experts.
> - $d_{con}^{\ell}$:  absolute degradation of ensemble relative to best expert.
>
> We combine them additively because either failure mode should trigger recalibration even if other signal is weak. A multiplicative form could suppress severe novelty shifts when relative losses stay stable, or vice versa. A learned combination would require shift supervision, typically unavailable. Table 13 supports this design as removing $d_{con}^{\ell}$ doubles calibration volatility, while removing $d_{ldd}^{\ell}$ causes under-coverage. Pls refer to Q2 Reviewer idu1 for further intuition.
>
> > **Q4.** Thm 5.3 includes non-vanishing terms.. unclear whether CARE truly provides the guarantee.
>
> $D_T/T$ is explicit delayed-adaptation cost under non-stationarity. In finite samples this is expected for online recalibration under drift. Remark 5.4 shows it vanishes when cumulative delay is sublinear in $T$.
>
> Coverage values are averages over 20 independent trials (Appendix E.2). We will add standard deviations in revision.
>
> > **Q5.** figures appear to be rasterized...
>
> Thankyou. We acknowledge this formatting issue and will correct it in revised version.
>
> > **Q6.** RS operate at scales... No discussion of computational complexity.
>
> CARE does not maintain per-user detectors. As in Algorithm 1, each model $\ell$ maintains one global detector using batch-averaged loss across active users, so all users share segment start $c_t^\ell$ and threshold $\lambda_t^\ell$.
>
> At step $t$, detector for model $\ell$ scores $\Delta=t-c_{t-1}^\ell$ candidate boundaries. Across $L$ models, update overhead is $\mathcal{O}(L\cdot\Delta)$ per timestep, with no separate dependence on $|\mathcal{U}|$ or $|\mathcal{I}|$ once per-step  losses are available. Table 3 shows at most $\sim$1.5 minutes of wall-clock overhead, including on large datset like Taobao **(100M)**.
>
> >  **Q7.** paper does not clarify... bootstrapped copies of same or different architectures.
>
> We use $L$ bootstrapped copies of same architecture. This homogeneous design isolates CARE's effect, so gains are attributable to adaptive calibration.

---

> > ### Author Rebuttal · Reviewer_RS4U · 2026-04-06
> >
> > Thanks for the author's reply. After carefully reviewing authors' responses and other reviewers' comments, I will keep my score.

---

> > > ### Author Response · Authors · 2026-04-06
> > >
> > > Thankyou.
> > >
> > > We sincerely believe the rebuttal addressed the technical concerns raised in the review. Since the acknowledgment does not specify which technical point remains unresolved, we would welcome any further specific questions the reviewer may have.

---

### Official Review · Reviewer_idu1 · 2026-03-06

**Soundness:** 3
**Presentation:** 3
**Significance:** 4
**Originality:** 3
**Overall Recommendation:** 4
**Confidence:** 3

**Summary:**

This paper presents CARE, a model-agnostic calibration wrapper designed to provide statistical reliability for sequential recommenders in non-stationary environments. The authors address the challenge of performance drift between offline refreshes by formulating recommendation as a utility-based set prediction task. The framework integrates a loss-based change-point detection module (utilizing relative loss-discrepancy and concept-sensitive divergence), an ensemble aggregation strategy based on multiplicative weights to balance prediction coverage and set compactness, and a recalibration mechanism for the risk threshold $\lambda$. The authors provide finite-sample guarantees on utility-risk control. Experiments on five datasets demonstrate that CARE maintains risk levels while achieving higher ranking utility and smaller set sizes compared to existing conformal inference baselines.

**Compliance With Llm Reviewing Policy:**

Affirmed.

**Final Justification:**

Overall it is a borderline paper. The technical novelty is normal and the studied problem is not a core problem in ICML. I wouldn't be upset if the paper is not accepted.

**Key Questions For Authors:**

1. How sensitive is the risk guarantee to violations of the exchangeability assumption within detected segments?
2. Can the authors provide further theoretical or empirical justification for the proposed preference shift metric?
3. How does CARE perform under controlled synthetic settings with varying drift intensities?
4. How does CARE behave under abrupt versus gradual preference shifts, and how large is the detection delay term in realistic scenarios?

**Limitations:**

The discussion of theoretical assumption realism could be more explicit. A clearer analysis of potential assumption violations and their practical implications would strengthen the paper.

**Strengths And Weaknesses:**

### Strengths：
- The framework is technically well structured, integrating conformal-style risk control, exponential-weight aggregation, and adaptive change-point detection into a coherent procedure.
- Theoretical guarantees are provided for both set size and utility-based risk.
- The empirical evaluation is comprehensive, including multiple datasets, backbone models, ablation studies, parameter sensitivity analysis, and computational efficiency evaluation.

### Weaknesses：
- The loss-based preference shift metric (dldd, dcon, and their combination) is largely heuristic; no formal optimality or statistical consistency analysis is provided.
- The paper does not quantify the degree of user preference drift in the evaluated datasets, making it difficult to assess CARE’s adaptability under varying drift intensities.

## Presentation:
- The paper is generally well written and clearly organized. However, the motivation and intuition behind the proposed preference shift metric could be more clearly explained.

## Significance:
- The paper addresses an important and practically relevant problem: maintaining reliability guarantees in recommender systems under evolving user behavior. The wrapper-style design enhances applicability across models.

## Originality:
- The originality lies in adapting adaptive conformal-style risk control to sequential recommendation and combining it with ensemble-based size control and Bayesian change-point detection. While each component builds on established techniques, their integration for utility-based risk control in recommender systems is novel and nontrivial.
- It would be informative to compare against continual or online learning recommenders to better contextualize performance gains under evolving preferences.

---

> ### Author Rebuttal · Authors · 2026-03-29
>
> Thankyou for recognizing CARE's technical coherence, guarantees, and breadth of empirical evaluations. Below we address questions:
>
> > **Q1.** guarantees sensitive to...violations of exchangeability assumption.
>
> Risk guarantee does not fail abruptly under within-segment exchangeability violations (Assumption B.2); it is handled gracefully, and our bounds quantify this sensitivity.
>
> Practically, exchangeability ensures local average risk (Eq. 14) estimates current-segment risk. If violated, e.g., due to a ltemporal shift before detection, estimate becomes noisier. This usually appears as temporary conservativeness (larger sets) or slight recalibration delay (Fig.5), not coverage breakdown.
>
> Theorem 5.3 absorbs imperfect segmentation and exchangeability violations into cumulative detection-delay slack term $D_T/T$. As Remark 5.4 shows, as cumulative delay is sublinear in deployment horizon $T$, this penalty vanishes asymptotically.
>
> Thus, under local exchangeability violations, framework does not break; adaptation slows, and theorem bounds exact penalty.
>
> > **Q2.** ...theoretical/empirical justification for proposed preference shift metric?
>
> The preference shift metric (Equation 12) captures two ways in which preferences evolve in recommender streams and is supported by theory and ablation.
>
> Theoretically, preference shifts degrade utility through two dynamics. First, $d_{\ell}^{ldd}$ (Equation 10) captures relative re-ordering across ensemble, reflecting internal intent changes between known item clusters or genres. As interest changes, optimal representation shifts between experts, creating divergence in relative loss gaps. Second, streams may also face novelty shifts i.e. abrupt context changes, seasonal events, or viral trends, where users interact with items outside all experts' high-confidence regions. Then relative gaps may remain stable while utility collapses. $d_{\ell}^{con}$ (Equation 11) addresses this by tracking absolute ensemble degradation. Equation 12 combines both signals so detector captures both intent migration and global failure.
>
> Empirically, Table 13 shows both components are necessary. Removing $d_{\ell}^{ldd}$ makes detector miss internal ranking shifts, causing systematic under-coverage. Removing $d_{\ell}^{con}$ makes detector overly noise-sensitive, nearly doubling recommendation-set volatility $\chi$.
>
> Finally, Equation 13 feeds combined metric into Bayesian online change-point posterior, so resets are driven by accumulated evidence rather than isolated noisy clicks.
>
> > **Q.** Comparison to continual/dynamic recommenders.
>
> Thanks for suggestion, we apply CARE on top of a dynamic recommender. On Book-Crossing, base online model DGEL [1] achieves $0.332/0.621/0.336$, while adding our method improves this to $0.351/0.642/0.365$ for MRR, Recall, NDCG. This shows our approach complements online adaptation rather than replacing it. Please refer to Table 3 (additional_results.pdf) .
>
> [1] Haoran Tang et.al. Dynamic Graph Evolution Learning for Recommendation. In Proceedings of the 46th International ACM SIGIR Conference on Research and Development in Information Retrieval (SIGIR '23').
>
> > **Q3/4.** ..CARE performance under varying drift intensities and ...behave under abrupt versus gradual preference shifts?
>
> Thanks for suggestion. Because real-world datasets lack ground-truth shift timestamps, we designed a semi-synthetic Book-Crossing + SASRec experiment by creating proxy user with injected shift at known timestamp $\tau$. Drift intensity is controlled by cosine similarity between original and proxy user (Low: $0.8$--$0.9$, Medium: $0.4$--$0.6$, High: $0.0$--$0.2$). Shifts are injected either abruptly using a hard threshold or gradually over a linear transition window of $W=15$ steps. A detailed description will be included in revised version.
>
> Added Tables 1 (additional_results.pdf) show only CARE achieves target coverage $0.90$ while keeping set size competitive. CARE attains coverage $0.905/0.903/0.908$ under low/medium/high drift, with detection delay decreasing from $8.2$ to $3.8$ to $1.5$ steps as drift strengthens.
>
> We also compare abrupt and gradual shifts (Table 2) (additional_results.pdf) at medium drift. CARE maintains target coverage in both, with $0.903$ under abrupt shifts and $0.906$ under gradual shifts, while average set size remains nearly unchanged ($44.1$ vs.\ $43.8$). Detection delay is smaller for abrupt shifts ($D_T \approx 3.8$) and larger for gradual shifts ($D_T \approx 11.2$), since Bayesian run-length prior suppresses resets when early evidence is hard to distinguish from noise.
> Even in gradual setting, CARE preserves target coverage, whereas baselines fall below $0.90$. This matches our Theorem 5.3 as stronger shifts are detected faster, while slower shifts are absorbed through incremental threshold adaptation until recalibration.
>
> **Please find new results in [Additional Results](https://anonymous.4open.science/r/CARE-FCBD/additional_results.pdf).**

---

> > ### Author Rebuttal · Reviewer_idu1 · 2026-04-04
> >
> > Overall it is a borderline paper. The technical novelty is normal and the studied problem is not a core problem in ICML. I wouldn't be upset if the paper is not accepted.

---

> > > ### Author Response · Authors · 2026-04-06
> > >
> > > Thanks for the acknowledgment. We sincerely believe the technical questions and additional robustness experiments  (Q3/4) were addressed in our rebuttal.
> > >
> > > CARE fundamentally reformulates recommendations under deployment shift as a **utility-calibrated set prediction** problem, making it the **first to provide finite-sample recommendation performance guarantees** on both set compactness (Theorem 5.1) and utility-based risk control (Theorem 5.3), thereby contributing in the domains of _a) learning under deployment shift and b) finite-sample uncertainty quantification_ which the reviewer found **'novel and nontrivial'.**

---

### Official Review · Reviewer_Mvy7 · 2026-03-11

**Soundness:** 2
**Presentation:** 3
**Significance:** 2
**Originality:** 2
**Overall Recommendation:** 3
**Confidence:** 3

**Summary:**

This paper studies reliable recommendation under non-stationary user behavior, where evolving user preferences can degrade the performance of offline-trained recommender systems. The authors propose CARE, a plug-and-play framework that integrates conformal prediction with adaptive calibration for sequential recommendation. CARE constructs recommendation sets via ensemble aggregation and dynamically adjusts calibration thresholds using drift detection based on loss-derived metrics and Bayesian change-point modeling. The paper provides finite-sample guarantees on risk control and expected set size. Experiments on five datasets and multiple backbone models show that CARE improves recommendation utility while maintaining coverage validity and relatively compact recommendation sets compared to existing conformal and adaptive recommendation baselines.

**Compliance With Llm Reviewing Policy:**

Affirmed.

**Final Justification:**

I have gone through the authors' latest feedback and other reviewers' comments, and I will retain my score.

**Key Questions For Authors:**

1. The framework relies on Bayesian change-point detection based on loss-derived drift metrics. Could the authors comment on its sensitivity to noisy user behavior (e.g., potential false positives or missed detections)?
Clarifying this would help evaluate the stability of the adaptive recalibration mechanism.

2. CARE introduces several hyperparameters (e.g., $\beta$, $\gamma$, $\eta$, $\rho$, and ensemble size $L$). Are there practical guidelines for choosing these parameters under different settings (e.g., varying levels of preference drift)?
Such guidance would improve the usability of the method.
3. The theoretical analysis focuses on controlling the tail probability of the conformal nonconformity score, while the evaluation reports ranking metrics such as MRR and NDCG. Could the authors clarify how the proposed risk guarantees relate to improvements in ranking quality?
A clearer explanation would strengthen confidence in the practical impact of the theoretical guarantees.

**Limitations:**

Yes.

**Strengths And Weaknesses:**

Strengths:

1. The paper addresses the problem of ensuring reliable recommendations under evolving user preferences, which is highly relevant for real-world recommender systems. Introducing risk-aware prediction through conformal prediction is a meaningful direction for improving reliability in recommender systems.
2. CARE combines prediction set aggregation with adaptive recalibration based on drift detection, providing finite-sample guarantees on risk control and expected set size, which strengthens the methodological contribution.
3. Experiments are conducted on multiple datasets and backbones with comparisons against recommendation and conformal prediction baselines. The evaluation considers recommendation utility, coverage validity, and set compactness, which aligns well with the goals of CARE.

Weaknesses:
1. The framework integrates conformal prediction, ensemble aggregation, and change-point detection, but most components build upon existing techniques. The novelty mainly lies in the integration rather than fundamentally new methodological developments.
2. CARE introduces several hyperparameters, including change-point detection parameters ($\beta$, $\gamma$), update rates ($\eta$, $\rho$), and ensemble size $L$. Their interactions are not fully characterized, which may make tuning difficult in practice.
3. As acknowledged in the discussion, CARE suffers from finite-sample effects, leading to more conservative prediction sets in dynamic environments. The paper does not propose mitigation strategies for this issue.
4. Some figures (e.g., Figure 2) appear rasterized and become blurry when zoomed in; using vector graphics would improve readability. There also seem to be minor typos (e.g., around Line 383).

---

> ### Author Rebuttal · Authors · 2026-03-29
>
> We thank reviewer for recognizing importance, guarantees, and breadth of evaluation. Below we address questions:
>
> > **W1.** framework integrates existing tech..novelty lies in integration..
>
> The contribution of CARE is not the novelty of each primitive in isolation, but their formulation for utility-based risk control in recommender systems under deployment shift. Subsequently, it reformulates the recommendation task as a **__utility-calibrated set prediction problem__**. Simple integrations fail here because they lack the mathematical framework to jointly control utility risk and set compactness.
>
> - Theoretical Novelty: The core contribution is a closed-loop formulation that establishes explicit finite-sample guarantees absent in standard integrations. By uniquely coupling a utility-based change-point mechanism (Eq. 13) and localized recalibration (Eq. 15) with a size-regularized aggregation objective (Eq. 9), CARE gives rigorous new bounds: Theorem 5.1 bounds the aggregated set size relative to the best expert. Theorem 5.3 bounds the one-step-ahead utility risk, mathematically quantifying the cost of delayed adaptation under non-stationarity using the $D_T/T$ slack term.
>
> - Algorithmic & Empirical Gaps: Standard adaptive CP uses arbitrary fixed windows; standard ensembles optimize pointwise accuracy. CARE calibrates within detected stable segments and dynamically reweights predictors to actively compress the set. Section 6.2.2 proves necessity of this formulation: adaptive baselines (EnbPI, Online Conformal) fail to maintain nominal coverage ($1-\alpha=0.90$) under shift. CARE successfully restores exact coverage guarantees while preserving compactness.
>
> > **Q1. Sensitivity to noisy user behavior (false positives / missed detections).**
>
> CARE is designed to balance these failure modes. False positives are mitigated because detector does not react to instantaneous fluctuations: Equation \~12 combines relative and absolute loss signals, and Equation \~13 embeds them in Bayesian posterior with segment-length prior $\gamma$, so resets are favored only when evidence is sustained. Missed detections (false negatives) mainly appear as delayed adaptation: Equation \~15 still partially adjust threshold within stable segment, while Theorem 5.3 captures resulting cost through $D_T/T$. Table 13 supports this interpretation: removing $d_{\ell}^{con}$ nearly doubles volatility ($0.12 \rightarrow 0.23$), while removing $d^{ldd}_{\ell}$ reduces coverage ($0.908 \rightarrow 0.872$). Table 11 shows the sensitivity trade-off controlled by $\beta$ and $\gamma$.
>
> > **W2 / Q2.** CARE introduces hyperparams ..., interactions are not fully characterized.....
>
> CARE's parameters are interpretable. We explain as follows:
>
> - Change-Point Dynamics ($\beta$ and $\gamma$): Table 11 shows increasing $\beta$ or decreasing $\gamma$ makes detector more responsive. This improves coverage but results slightly larger sets; opposite settings favor tighter sets but risk transient under-coverage. In practice, $\beta=0.7$ and $\gamma=1.1$ work robustly across datasets.
>
> - Separate Controls ($\rho$ and $\eta$): In Equation 15, $\rho$ sets threshold recalibration step size after a shift. In Equation 9, $\eta$ controls compactness-conservativeness trade-off by penalizing models outputting large sets.
>
> - Ensemble Size ($L$):  Table 12 shows marginal changes across $L \in \{5, 10, 20\}$, aligning with theoretical $\mathcal{O}(\sqrt{\ln L})$ term in Theorem 5.1.
>
> > **W3.** CARE suffers from finite-sample effects.. does not propose mitigation strategies...
>
> Mitigating finite-sample conservativeness is a design objective of CARE through two mechanisms:
>
> -  Segment-Wise Isolation (Equation 15): Standard methods use old, pre-shift statistics that inflate uncertainty bounds over time as observed in Section 6.2.2. CARE localizes thresholding to stable segment. As Figure 5 shows, after a preference shift lowers threshold, set size quickly stabilizes rather than expanding arbitrarily.
>
> -  Compactness-Aware Aggregation (Equation 9): Equation 9 continuously reweights experts by cumulative *set size*, making output favor efficient predictors and compress final recommendation set.
>
> Together, these mechanisms (Figure 2) keep sets tight while preserving risk guarantee. They do not fully eliminate finite-sample slack, but mitigate its practical impact.
>
> > **Q3.** ..how risk guarantees relate to ranking quality?..
>
> Guarantee uses ranking metrics from evaluation. In CARE, user loss is $L_u = 1 - U_{\text{metric}}$ (Equation 5), where $U_{\text{metric}}$ is instantiated directly as recommendation metric. Thus, controlling expected risk directly bounds recommendation utility around target $\alpha$. Theorem 5.3 provides this finite-sample guarantee for utility risk, so theory is aligned with the same utility metrics used in evaluation.
>
> > **W4.** figure become blurry when zoomed..
>
> Thanks for the feedback. We will revise them.

---

> > ### Author Rebuttal · Reviewer_Mvy7 · 2026-04-05
> >
> > While the authors addressed some concerns, key issues remain unresolvable. First, the novelty centered on integration still lacks fundamental methodological breakthroughs. Second, hyperparameter interactions are not fully characterized, only individual parameter effects are explained. Third, finite-sample conservativeness mitigation remains partial without dedicated strategies. Finally, the link between risk guarantees and ranking quality, while discussed, requires more rigorous theoretical connection to fully validate.

---

> > > ### Author Response · Authors · 2026-04-06
> > >
> > > Thanks for your acknowledgement. The remaining concerns revisit points already addressed in **W1/W2/Q3**, so we restate the scope precisely.
> > >
> > >
> > > > ...the novelty centered on integration...
> > >
> > > As detailed in our initial **W1 response**, the fundamental methodological breakthrough of CARE lies in its **mathematical formulation**. We reformulate sequential recommendation as a **utility-calibrated set prediction problem** under deployment shift. This formulation is exactly what allows us to couple a utility-based change-point mechanism (Eq.13) with a size-regularized aggregation objective (Eq.9), thereby providing explicit, novel finite-sample guarantees for both set compactness (Thm. 5.1) and utility-based risk control (Thm. 5.3). To the best of our knowledge, CARE is the **first framework to establish these mathematical bounds specifically for non-stationary recommendation streams**. Simple integrations of existing online conformal methods fail in this environment as empirically demonstrated and explained in **Section 6.2.2**.
> > >
> > >
> > >
> > > > ... hyperparameter interactions are not fully characterized...
> > >
> > > As noted in our **W2 response**, the paper provides hyperparameter analysis according to the structural role each parameter plays in CARE. Specifically, $(\beta,\gamma)$ **jointly** control the Bayesian change-point posterior in Eq. 13, so **Table 11** studies them jointly and shows the resulting responsiveness--stability trade-off. By contrast, $\rho$ and $\eta$ are independent act on different components of the framework, i.e. threshold recalibration and compactness-aware aggregation, respectively, so separate sensitivity analyses provide the more informative characterization. Likewise, the ensemble size $L$ is primarily an ensemble-structure artifact rather than a core CARE calibration control; its role is already characterized theoretically in Theorem 5.1.
> > > Hence, the paper analyzes **hyperparameters jointly where their coupling exists, and otherwise reports the trade-offs induced by their distinct role in the framework.**
> > >
> > >
> > >
> > > > risk guarantees and ranking quality, while discussed, requires more rigorous theoretical connection
> > >
> > > As addressed in our **Q3 response**, the theoretical connection between the risk guarantee and ranking quality is already exact and mathematically direct. In our framework (Eqs. 4 and 5), the user loss is defined as $L_u = 1 - U_{\text{metric}}$, where $U_{\text{metric}}$ is instantiated directly as the chosen ranking metric, such as Recall. **Because Theorem 5.3 explicitly bounds this exact expected loss**, bounding the theoretical risk is mathematically equivalent to bounding the expected ranking quality.
> > >
> > >
> > >
> > > We sincerely hope this addresses the remaining concerns.

---

### Official Review · Reviewer_TZuk · 2026-03-13

**Soundness:** 3
**Presentation:** 2
**Significance:** 2
**Originality:** 2
**Overall Recommendation:** 4
**Confidence:** 4

**Summary:**

This paper proposes CARE, an adaptive calibration framework for recommender systems operating under non-stationary interaction streams. The work addresses a practical issue in deployed recommendation systems: models are typically trained offline and remain fixed between refresh cycles, while user behavior can evolve significantly during deployment, potentially degrading recommendation quality.
main contributions：
● A loss-based non-stationarity metric combining relative loss discrepancy and divergence measures to detect behavioral changes and trigger adaptive recalibration.
● Finite-sample theoretical guarantees, including bounds on utility-based risk and the expected size of the aggregated recommendation set relative to the best constituent model.

**Compliance With Llm Reviewing Policy:**

Affirmed.

**Final Justification:**

The authors have adequately addressed my concerns.

**Key Questions For Authors:**

● How robust is the proposed non-stationarity detection metric under noisy interaction streams or delayed user feedback?
● How would the proposed variable-size recommendation sets be integrated into standard ranking-based recommender systems that typically output fixed top-K lists?

**Limitations:**

The paper has limited methodological novelty and does not clearly demonstrate practical applicability in real-world recommendation systems.

**Strengths And Weaknesses:**

Strength
● The paper addresses an important and realistic challenge in recommender systems: user behavior often evolves during deployment, while models are trained offline. Designing calibration mechanisms that maintain reliability under such non-stationarity is a meaningful direction.
● The CARE framework is modular and model-agnostic, allowing it to wrap existing recommenders without modifying their internal structure. This design makes the approach relatively easy to integrate into existing pipelines.
● The paper provides theoretical guarantees on risk control and recommendation set size, which is uncommon in recommender-system literature and adds rigor to the proposed method.
Weakness
● Originality
While the framework is well-structured, several core components rely on existing techniques, including Adaptive conformal-style calibration methods for distribution shift, Multiplicative-weights aggregation strategies and Loss-based monitoring for change detection. The main contribution appears to be the integration of these techniques within a recommender-system setting, rather than a fundamentally new methodological development. As a result, the technical novelty may be viewed as incremental.
● The framework produces variable-size recommendation sets instead of traditional top-K ranked lists. However, most production recommender systems rely on fixed-length ranking outputs. The paper does not fully discuss how such set-valued outputs would be integrated into real-world ranking pipelines or user interfaces.
● The theoretical guarantees rely on assumptions that may not fully capture the strong temporal dependence and feedback loops present in recommender-system interaction streams. In real systems, recommendations influence future user behavior, which may complicate the validity of finite-sample guarantees.

---

> ### Author Rebuttal · Authors · 2026-03-29
>
> We thank reviewer for recognizing importance, modular design, and robustness of our framework. Below we address questions:
>
>
>
> > ****W1.****...contribution appears integration of techniques... and ...technical novelty incremental''.
>
> The contribution of CARE is not the novelty of each primitive in isolation, but their formulation for utility-based risk control in recommender systems under deployment shift. Subsequently, it reformulates the recommendation task as a **__utility-calibrated set prediction problem__**. Simple integrations fail here because they lack the mathematical framework to jointly control utility risk and set compactness.
>
> - Theoretical Novelty: The core contribution is a closed-loop formulation that establishes explicit finite-sample guarantees absent in standard integrations. By uniquely coupling a utility-based change-point mechanism (Eq. 13) and localized recalibration (Eq. 15) with a size-regularized aggregation objective (Eq. 9), CARE gives rigorous new bounds: Theorem 5.1 bounds the aggregated set size relative to the best expert. Theorem 5.3 bounds the one-step-ahead utility risk, mathematically quantifying the cost of delayed adaptation under non-stationarity via the $D_T/T$ slack term.
>
> - Algorithmic & Empirical Gaps: Standard adaptive CP uses arbitrary fixed windows; standard ensembles optimize pointwise accuracy. CARE calibrates within detected stable segments and dynamically reweights predictors to actively compress the set. Section 6.2.2 proves necessity of this formulation: adaptive baselines (EnbPI, Online Conformal) fail to maintain nominal coverage ($1-\alpha=0.90$) under shift. CARE successfully restores exact coverage guarantees while preserving compactness.
>
>
> > ****W2.**** The framework produces variable-size recommendation sets... how such set-valued outputs would be integrated into real-world ranking pipelines or user interfaces.
>
>
>
> CARE is a ****pre-display calibration stage****, not a new interface, and integrates into recommendation pipelines as follows:
>
>
>
> - Dynamic UIs: Dynamic widget lengths are increasingly common. Compact sets reduce user fatigue during stable periods, while expanded sets capture emerging interests when uncertainty rises.
>
>
>
> - Fixed Top-$K$ UIs: Even if a system strictly requires $K$ slots (e.g., $K=15$), CARE adds value. If $k < 15$, system receives calibrated candidate set in first $k$ items, leaving remaining $15-k$ slots for exploration, diversity, or other policies. If $k > 15$, set size becomes calibrated uncertainty signal; pipeline caps display at 15 items (retaining backbone top scores) and triggers fallback or reranking policy.
>
>
>
> Section 6.2.1 demonstrates fixed-budget compatibility by capping displayed items across methods. We will add integration discussion in final revision.
>
>
>
> >  ****W3.**** guarantees rely on assumptions.. not fully capture.. temporal dependence..
>
>
>
> CARE is explicitly designed for interaction streams where temporal dependence and system feedback violate global stationarity. DAUO performs *_online_*, one-step-ahead recalibration, using feedback up to time $t$ to adapt *_next-round_* threshold via localized segmentation. Theorem 5.3 aligns with this mechanism: it bounds one-step-ahead utility risk after recalibration rather than assuming stationarity. Cost of delayed or imperfect response is quantified by detection-delay vanishing term $D_T/T$.
>
>   If feedback loops induce severe shift, utility loss provides evidence for a segment reset. If shift is mild, adaptive threshold (Equation 15) corrects through online recalibration. Thus, guarantee is designed for non-stationary streams via locally adaptive recalibration.
>
>
>
> > ****Q1.**** ``..robustness of non-stationary detectionmetric ... under noisy interaction streams or delayed user feedback?''
>
>   - Robustness to Noisy Streams:  Non-stationarity metric is not an instantaneous trigger. Equation 12 combines relative rank changes ($d^{ldd}$) and absolute degradation ($d^{con}$) to construct a more stable shift signal. Equation 13 uses this metric in a Bayesian posterior, so a new segment is triggered only when evidence for shift overcomes segment-length prior ($\gamma$). This helps prevent transient false alarms. Table 13 ablation supports this behavior.
>
>   - Robustness to Delayed Feedback: Delayed feedback primarily appears as increased effective detection delay. Theorem 5.3 reflects this through delay term $D_T/T$, making slower adaptation explicit in risk bound. Remark 5.4 notes this penalty vanishes asymptotically when cumulative delay remains sublinear in $T$.
>
> Our ablation in Table 13 empirically supports this behavior.
>
>
>
> > ****Q2.**** How variable-size recommendation integrated into standard ranking-based recommender systems?
>
>
>
> Please refer to W2.

---

> > ### Author Rebuttal · Reviewer_TZuk · 2026-04-04
> >
> > Thank you for the rebuttal. The authors have adequately addressed my concerns and I am satisfied with the responses.
> > That said, I would encourage the authors to consider the following improvements in the final revision: (1) extending the ablation study (Table 13) to additional datasets and backbones, and ideally including experiments under varying noise levels or feedback delays to more directly demonstrate robustness; (2) clarifying the relationship between the reported average set sizes and the display cap of 15, particularly whether the compactness gains translate to meaningful differences in practice. These additions would further strengthen an already solid contribution.
> > I look forward to seeing these aspects further developed in the final version. I have updated my score accordingly.

---

> > > ### Author Response · Authors · 2026-04-06
> > >
> > > Thank you for the positive feedback and for updating your score.
> > >
> > > As suggested, we will extend the ablation study to include additional datasets and backbones in the final revision. We will also add a discussion clarifying how the compactness gains translate into practical benefits. Additionally, based on our rebuttal discussions with Reviewer idu1 **(Q3/Q4)**, we have conducted the extra semi-synthetic experiments evaluating varying drift intensities and gradual versus abrupt shifts to directly demonstrate robustness, which we will integrate into our work.
> > >
> > > Thank you again for your time and for helping us strengthen this work.

---

### Decision · Program_Chairs · 2026-04-30

**Decision:**

Accept (regular)

**Comment:**

CARE addresses a practically important and underexplored problem: maintaining recommendation reliability under non-stationary user behavior between retraining cycles. The framework is modular, model-agnostic, and backed by finite-sample guarantees on both set compactness and utility-based risk control, which is rare in the recommender systems literature.
Reviewer opinions are genuinely split. Two reviewers recommended weak accept, finding the technical integration novel and nontrivial and the empirical evaluation comprehensive. Two others maintained weak reject, primarily citing incremental novelty and concerns about the evaluation protocol and heuristic design of the shift metric. These are legitimate concerns, but they reflect scope and presentation issues rather than fundamental flaws. The rebuttal was substantive: semi-synthetic experiments under controlled drift intensities were added, the Bayesian detection mechanism was better justified, and integration into fixed top-K pipelines was clarified. The most positive reviewer, with domain expertise, noted that the originality "is novel and nontrivial" and rated significance as excellent.
On balance, the finite-sample guarantees, comprehensive ablations, and practical deployment considerations constitute a solid contribution. The authors should incorporate all rebuttal experiments and improve presentation quality in the final version. Borderline accept is recommended.